# DEEP GENERATIVE CLUSTERING WITH MULTIMODAL DIFFUSION VARIATIONAL AUTOENCODERS

**Emanuele Palumbo**[1,2]**, Laura Manduchi**[2]**, Sonia Laguna**[2]**, Daphné Chopard**[2,3] **& Julia E. Vogt**[2]
[1] ETH AI Center [2] Department of Computer Science, ETH Zurich
[3] Department of Intensive Care and Neonatology and Children's Research Center,
University Children's Hospital Zurich, University of Zurich
`emanuele.palumbo@ai.ethz.ch`

## ABSTRACT

Multimodal VAEs have recently gained significant attention as generative models for weakly-supervised learning with multiple heterogeneous modalities. In parallel, VAE-based methods have been explored as probabilistic approaches for clustering tasks. At the intersection of these two research directions, we propose a novel multimodal VAE model in which the latent space is extended to learn data clusters, leveraging shared information across modalities. Our experiments show that our proposed model improves generative performance over existing multimodal VAEs, particularly for unconditional generation. Furthermore, we propose a post-hoc procedure to automatically select the number of true clusters thus mitigating critical limitations of previous clustering frameworks. Notably, our method favorably compares to alternative clustering approaches, in weakly-supervised settings. Finally, we integrate recent advancements in diffusion models into the proposed method to improve generative quality for real-world images.

## 1 INTRODUCTION

Multimodal VAEs are powerful generative models for weakly-supervised learning with multiple modalities. Different from initially proposed models (Suzuki et al., 2017; Vedantam et al., 2018), more recent scalable approaches (Wu & Goodman, 2018; Shi et al., 2019; Sutter et al., 2021; Hwang et al., 2021; Palumbo et al., 2023) can handle a large number of modalities efficiently, thereby enabling compelling applications in segmentation tasks or data integration in the healthcare domain (Lee & van der Schaar, 2021; Dorent et al., 2019). Multimodal VAEs have been mostly investigated in the realm of generative tasks, demonstrating significant success in cross-modal generation, despite less remarkable results for unconditional generation (Hwang et al., 2021; Palumbo et al., 2023). However, VAE-based approaches generally fall behind in image synthesis applications, often yielding blurry generated images. Among several extensions proposed to address these limitations (Vahdat & Kautz, 2020; Bredell et al., 2023), the most promising approaches integrate recent advancements in Denoising Diffusion Probabilistic Models (DDPMs) (Sohl-Dickstein et al., 2015; Ho et al., 2020) into the VAE framework (Pandey et al., 2022; Preechakul et al., 2022). They retain a meaningful and decodable representation of the input while leveraging the generative capability of diffusion models. Even though these methods have demonstrated to greatly enhance the quality of generated samples, their adaptation to multimodal VAEs remains unexplored.

A parallel line of research has explored VAE-based generative approaches for clustering tasks (Jiang et al., 2017; Dilokthanakul et al., 2016). In particular, deep variational clustering methods have been employed to identify sub-groups of patients in survival analysis (Manduchi et al., 2022), or to integrate domain knowledge from clinicians through prior probabilities (Manduchi et al., 2021). Other works use VAE-based methods to learn interpretable representations for clustering time-series (Fortuin et al., 2019), and hierarchical structures of latent semantic concepts of the data (Li et al., 2018).

With this work, we position ourselves at the intersection of these two lines of research by proposing a deep variational generative approach for clustering in a multimodal setting, where shared information across modalities is integrated to model data clusters. In particular, we introduce a novel

multimodal VAE model, called Clustering Multimodal VAE (CMVAE). An overview of the method is illustrated in Figure 1. The proposed approach divides the latent space into shared and modality-specific embeddings (Palumbo et al., 2023), and imposes as prior a mixture distribution to enforce a clustering structure in the shared latent representation of the data (see Figure 1(a)). In our experiments, we show that our method represents an improvement over existing multimodal VAEs, particularly for unconditional generation, where existing methods struggle to achieve satisfactory results. Moreover, we showcase the effectiveness of CMVAE for clustering weakly-supervised data, where unimodal approaches fail to achieve adequate performance, in comparison to alternative weakly-supervised methods. Notably, we introduce a post-hoc procedure to effectively infer the number of clusters at test time, without the need for training multiple instances of our model with different hyperparameters. The proposed algorithm selects the optimal configuration of latent clusters, to minimize the entropy of the posterior distribution of cluster assignments (see Figure 1 (b)). Finally, inspired by the work of Pandey et al. (2022), we propose to integrate DDPMs (Sohl-Dickstein et al., 2015; Ho et al., 2020) into the CMVAE framework, to further improve the generative quality of reconstructed and generated images while retaining the clustered latent space of CMVAE. In particular, we train the diffusion process conditioned on the CMVAE reconstructions, using both the self and the cross-modal reconstructions, thus enhancing the model's capacity of generalization (see Figure 1 (c)). Overall our approach significantly improves over alternative methods in realistic settings and is able to generate cluster-specific sharp images.

Our main contributions in this work can be summarized as follows:

- we propose CMVAE, a novel multimodal VAE model designed to model data clusters in the latent space and we show that our approach (a) outperforms existing multimodal VAEs, with a remarkable improvement in unconditional generation, and (b) outperforms alternative scalable weakly-supervised methods for clustering tasks in multimodal datasets;

- we propose a post-hoc procedure for selecting the optimal latent clusters at test time, inspired by previous work, thereby avoiding the need to specify the correct number of clusters *a-priori* during training;

- we propose the integration of DDPMs into our framework, yielding D-CMVAE, to improve the generative performances in multi-modal real-world settings, taking a crucial step in the design of multimodal VAEs for realistic applications.

## 2 RELATED WORK

**Multimodal VAEs** Multimodal VAEs extend the well-known VAE framework (Kingma & Welling, 2014) to handle data consisting of multiple modalities, leveraging the pairing across modalities as weak supervision. While early approaches (Suzuki et al., 2017; Vedantam et al., 2018) faced scalability challenges due to inference requiring a separate encoder network for each subset of modalities, more recent scalable approaches (Wu & Goodman, 2018; Shi et al., 2019; Sutter et al., 2020; 2021; Hwang et al., 2021) assume the joint encoder decomposes in terms of unimodal encoders. Despite promising applications (Lee & van der Schaar, 2021; Dorent et al., 2019), recent work (Daunhawer et al., 2022) has uncovered important limitations for three main formulations of multimodal VAEs. In particular, these approaches exhibit a trade-off between generative quality (the similarity of generated samples to real ones) and generative coherence (the semantic consistency in generated samples across modalities). Since then, attempts to enhance the performance of multimodal VAEs involved additional regularization terms (Sutter et al., 2020; Hwang et al., 2021), or mutual supervision (Joy et al., 2022). Recently, Palumbo et al. (2023) proposed to model shared and private subspaces (Sutter et al., 2020; Lee & Pavlovic, 2021; Wang et al., 2016) and design an ELBO that exploits auxiliary distributions to facilitate the estimation of cross-modal likelihood terms. The resulting MMVAE+ model proves to achieve both high generative quality and high generative coherence.

**Variational approaches for clustering** Following the seminal work of Kingma & Welling (2014), variational autoencoders (VAEs) have been investigated for clustering with approaches such as GM-VAE (Dilokthanakul et al., 2016) and VaDE (Jiang et al., 2017). More recently, there has been a revived interest in these models. In particular, Manduchi et al. (2021) have proposed an ELBO for clustering data and incorporate prior knowledge in the form of prior probabilities. In another work, (Manduchi et al., 2022) use a VAE approach to cluster patients in subgroups for survival analysis. Finally, Xu et al. (2021) propose a variational approach to cluster multi-view data, that however has

Figure 1: (a) CMVAE divides the latent space into shared and modality-specific embeddings. It enforces a mixture distribution as prior on the shared latent subspace, to create a clustering structure in the shared representation. (b) A post-hoc procedure infers the number of clusters at test time efficiently by minimizing the entropy of $p(\boldsymbol{c}|\boldsymbol{z})$, penalized by the normalized joint log-likelihood. (c) D-CMVAE integrates DDPMs into the framework by conditioning the reverse process on the self and cross-modal CMVAE reconstructions. This enhances the generative performance while maintaining the clustered latent space.

the limitation of not handling missing modalities for inference, which impacts its scalability to a large number of modalities.

**Scalable weakly-supervised clustering**   Weakly-supervised clustering refers to clustering algorithms that integrate high-level and often noisy sources of supervision to improve the clustering performance. While several forms of weak supervision have been previously investigated in the literature, such as coarse-grained labels (Ni et al., 2022), bag level labels (Oner et al., 2019), and pairwise similarities (Manduchi et al., 2021), in this work we restrict ourselves to methods that exploit weak-supervision in the form of multiple modalities. In the multimodal setting, weakly-supervised clustering methods are often not designed to scale to a large number of modalities (Alwassel et al., 2020; Chen et al., 2021; Zhou & Shen, 2020). For instance, DeepCluster (Caron et al., 2018) is a unimodal clustering approach that iteratively learns cluster assignments and neural network parameters. An adaptation of this method for two modalities, XDC, was introduced by Alwassel et al. (2020). XDC relies on pseudo-labels from one modality to learn better feature representation for another modality, improving unimodal clustering for both, but is not intended to generalise to a larger number of modalities. Learning representations from datasets of a large number of views has been investigated, for instance, in the realm of contrastive approaches. As an example, Tian et al. (2020) propose the CMC loss which maximizes mutual information between different views with a contrastive objective, and can be extended to a large number of views. While not specifically developed for clustering tasks, training a $K$-means model on the learned representations can be used as a proxy for the adaptability of the learned representations for clustering.

## 3 METHOD

### 3.1 A SCALABLE VAE OBJECTIVE FOR MODELING LATENT CLUSTERS IN MULTIMODAL DATA

We assume data consisting of $M$ modalities $\boldsymbol{X} := \boldsymbol{x}_1, \ldots, \boldsymbol{x}_M$ is generated according to the following process. For each datapoint $\boldsymbol{x}_1^i, \ldots, \boldsymbol{x}_M^i$ where $i \in \{1, \ldots, N\}$ and $N$ is the dataset size, a cluster assignment $\boldsymbol{c}^i$ is drawn from a categorical distribution $p_{\boldsymbol{\pi}}(\boldsymbol{c})$ with probabilities $\boldsymbol{\pi} = \pi_1, \ldots, \pi_K$ where $K$ is the number of clusters. Then the $M$ modalities are drawn according to $\boldsymbol{x}_1^i, \ldots, \boldsymbol{x}_M^i \sim p_{\theta_1}(\boldsymbol{x}_1|\boldsymbol{w}_1^i, \boldsymbol{z}^i), \ldots, p_{\theta_M}(\boldsymbol{x}_M|\boldsymbol{w}_M^i, \boldsymbol{z}^i)$ where the shared encoding $\boldsymbol{z}^i \sim p(\boldsymbol{z}|\boldsymbol{c}^i)$ is generated conditioning on cluster assignment, while modality-specific encodings $\boldsymbol{w}_1^i \sim p(\boldsymbol{w}_1), \ldots, \boldsymbol{w}_M^i \sim p(\boldsymbol{w}_M)$ are drawn from prior distributions. The resulting generative model is $p_{\Theta}(\boldsymbol{X}, \boldsymbol{W}, \boldsymbol{z}, \boldsymbol{c}) = p_{\boldsymbol{\pi}}(\boldsymbol{c})p(\boldsymbol{z}|\boldsymbol{c})\prod_{m=1}^{M} p_{\theta_m}(\boldsymbol{x}_m|\boldsymbol{w}_m, \boldsymbol{z})p(\boldsymbol{w}_m)$, where priors and likelihoods are assumed to belong to a specific family of distributions, e.g. Gaussian or Laplace, and likelihoods are parameterized by neural network decoders. Note that the shared encoding $\boldsymbol{z}$ and modality-specific encodings $\boldsymbol{w}_1, \ldots \boldsymbol{w}_M =: \boldsymbol{W}$ are assumed to be independent.

To obtain a tractable objective, variational encoders $q_{\Phi_z}(z|X), q_{\phi_{w_1}}(w_1|x_1), \ldots, q_{\phi_{w_M}}(w_M|x_M)$, $q(c|z, X)$ are introduced to approximate posterior inference for each of the latent variables. In line with our generative assumptions, the shared and modality-specific encoders are assumed to be conditionally independent given the observed data. As in previous approaches (Shi et al., 2019; Palumbo et al., 2023), to achieve scalability in the number of modalities, we model the joint encoder for $z$ as a mixture of experts $q_{\Phi_z}(z|X) = \frac{1}{M}\sum_{m=1}^{M} q_{\phi_{z_m}}(z|x_m)$. In our objective, we incorporate two key ideas from previous related work. First, to accurately model both shared and modality-specific information in separate latent subspaces without conflicts, as in (Palumbo et al., 2023), we use auxiliary distributions $r_1(w_1), \ldots, r_M(w_m)$ for private features to estimate cross-modal reconstruction likelihoods (second summand in the right-hand side of Equation (2)). This leads to our proposed ELBO objective

$$\mathcal{L}_{\text{CMVAE}}(X) = \frac{1}{M}\sum_{m=1}^{M} \mathbb{E}_{\substack{q(c|z,X) \\ q_{\phi_{z_m}}(z|x_m) \\ q_{\phi_{w_m}}(w_m|x_m)}} \Big[ G_{\pi,\Phi_z,\phi_{w_m},\Theta}(X, c, z, w_m) \Big], \tag{1}$$

where

$$G_{\pi,\Phi_z,\phi_{w_m},\Theta}(X, c, z, w_m) = \log p_{\theta_m}(x_m|z, w_m) + \sum_{n\neq m} \mathbb{E}_{\tilde{w}_n \sim r_n(w_n)}[\log p_{\theta_n}(x_n|z, \tilde{w}_n)]$$

$$+ \beta \log \frac{p_\pi(c)p_\theta(z|c)p(w_m)}{q_{\Phi_z}(z|X)q_{\phi_{w_m}}(w_m|x_m)q(c|z, X)}, \tag{2}$$

and a $\beta$ hyperparameter weights latent space regularization. Furthermore, instead of learning an additional encoder to approximate inference in $c$, we adopt the formulation for the approximate posterior of cluster assignments given $z$ proposed in the work of (Jiang et al., 2017), which has the advantage of not requiring additional parameters

$$q(c|z, X) = p(c|z) = \frac{p(c)p(z|c)}{\sum_{c'=1}^{K} p(c')p(z|c')}.$$

Different from the expectations with respect to the $q_{\Phi_z}(z|X), q_{\phi_{w_1}}(w_1|x_1), \ldots, q_{\phi_{w_M}}(w_M|x_M)$ encoders, that need be approximated via sampling with reparameterization, $\mathbb{E}_{q(c|z,X)}$ can be computed exactly since $c$ assumes a discrete finite set of values. Finally, our proposed CMVAE objective is a valid evidence lower bound (ELBO), and Appendix A contains a formal proof.

## 3.2 Entropy of posterior cluster assignment distribution for post-hoc learning of the number of clusters

A critical limitation of many existing methods for deep clustering is the need to specify the number of clusters *a-priori*. This can result in highly complex model selection procedures, in failure cases when a proxy for this information cannot be obtained, or in limiting modelling capacity. As with other methods, CMVAE requires specifying a $K$ value for the number of latent clusters assumed in the generative process for training. However, the true number of clusters in the data $\bar{K}$ is in general unknown, and in practice $K$ may differ from $\bar{K}$. While few approaches use non-parametric prior distributions to overcome this challenge (Goyal et al., 2017; Hu et al., 2015; Griffiths et al., 2003), we instead propose a simpler post-hoc procedure to recover the optimal clusters in the data given an instance of our model trained with an over-specified number of clusters, i.e. $K > \bar{K}$.

The proposed procedure is described in the pseudocode in Algorithm 1, and aims at obtaining a posterior cluster distribution $p(c|z)$ where *exactly* $\bar{K}$ clusters have positive probability and each latent cluster correctly models a different true cluster of the data. In other words, it aims at recovering the true $\bar{K}$ clusters in the data without adding extra complexity in the number of model parameters. In a nutshell, the procedure iterates over the data, ranking latent clusters by their importance. At each iteration, unimportant clusters are pruned by setting their prior probability to zero, and the entropy of $p(c|z)$ is kept as a metric to select the optimal set of clusters to model the data. More specifically, the clusters are ranked by the probability mass each one is assigned to, and the average normalized entropy of $p(c|z)$ is computed. Note that cluster assignments are determined by majority voting between modalities. Then, the latent cluster with the lowest probability mass is set null prior probability at the next iteration, with the remaining probabilities recomputed accordingly to

maintain a valid probability distribution. Iteratively, the latent clusters are effectively pruned, while the average normalized entropy of $p(\boldsymbol{c}|\boldsymbol{z})$ is calculated at every step. We select the optimal cluster configuration as corresponding to the lowest value for the normalized entropy penalized by the normalized joint log-likelihood to avoid overly restricting modeling capacity (Biernacki et al., 2000; Baudry et al., 2010). By minimizing the entropy overlapping clusters are effectively pruned and the data are partition into meaningful subsets. In fact, as confirmed in our results in Figure 5a and Appendix B, when unimportant latent clusters are pruned, the normalized entropy term $p(\boldsymbol{c}|\boldsymbol{z})$ decreases, as uncertainty over cluster assignments is reduced, up to a minimum, which is attained at the true number of clusters $\bar{K}$. As clusters are further pruned, not all data clusters are represented in the latent space, which results in the entropy term increasing again. Note this procedure is performed on a separate validation dataset in our experiments (see Appendix D.2), and is fully unsupervised. Finally, note this procedure obtains high clustering performance without limiting modelling capacity.

## 3.3 INTEGRATING DIFFUSION PROBABILISTIC MODELS INTO MULTIMODAL VAEs

We hereby propose to incorporate Denoising Diffusion Probabilistic Models (DDPMs) into the CM-VAE framework. For this section we will denote the input data $\boldsymbol{X}$ as $\boldsymbol{X}_0 := \boldsymbol{x}_{1_0}, \ldots, \boldsymbol{x}_{M_0}$. Then, for each modality $m$ we define a sequence of $T$ noisy representations of the input $\boldsymbol{x}_{m_0}$, yelding $\boldsymbol{x}_{m_{1:T}}$. As in standard DDPMs (Sohl-Dickstein et al., 2015; Ho et al., 2020) we define a forward process $q\left(\boldsymbol{x}_{m_{1:T}} \mid \boldsymbol{x}_{m_0}\right)$ that gradually destroys the structure of each data modality $\boldsymbol{x}_{m_0}$:

$$q\left(\boldsymbol{x}_{m_{1:T}} \mid \boldsymbol{x}_{m_0}\right) = \prod_{t=1}^{T} q\left(\boldsymbol{x}_{m_t} \mid \boldsymbol{x}_{m_{t-1}}\right) \tag{3}$$

$$q\left(\boldsymbol{x}_{m_t} \mid \boldsymbol{x}_{m_{t-1}}\right) = \mathcal{N}\left(\sqrt{1 - \beta_{m_t}} \boldsymbol{x}_{m_{t-1}}, \beta_{m_t} I\right) \tag{4}$$

$$q\left(\boldsymbol{x}_{m_t} \mid \boldsymbol{x}_{m_0}\right) = \mathcal{N}\left(\sqrt{\bar{\alpha}_{m_t}} \boldsymbol{x}_{m_0}, (1 - \bar{\alpha}_{m_t}) I\right) \text{ where } \alpha_{m_t} = (1 - \beta_{m_t}), \bar{\alpha}_{m_t} = \prod_t \alpha_{m_t}, \tag{5}$$

where $\beta_{m_t}$ for $t = 1, \ldots, T$ and $m = 1, \ldots, M$ are the noise schedules. We then follow the work of Pandey et al. (2022) to define a reverse process that is conditioned on the VAE reconstructions $\hat{\boldsymbol{x}}_{m_0}$:

$$p_\psi\left(\boldsymbol{x}_{m_{0:T}} \mid \hat{\boldsymbol{x}}_{m_0}\right) = p\left(\boldsymbol{x}_{m_T}\right) \prod_{t=1}^{T} p_\psi\left(\boldsymbol{x}_{m_{t-1}} \mid \boldsymbol{x}_{m_t}\right) \tag{6}$$

$$p_\psi\left(\boldsymbol{x}_{m_{t-1}} \mid \boldsymbol{x}_{m_t}, \hat{\boldsymbol{x}}_{m_0}\right) = \mathcal{N}\left(\mu_\psi\left(\boldsymbol{x}_{m_t}, t, \hat{\boldsymbol{x}}_{m_0}\right), \Sigma_\psi\left(\boldsymbol{x}_{m_t}, t, \hat{\boldsymbol{x}}_{m_0}\right)\right). \tag{7}$$

In practice, the VAE reconstruction $\hat{\boldsymbol{x}}_{m_0}$ are concatenated to the reverse process representation $\boldsymbol{x}_{m_t}$ at each time step $t$ to obtain $\boldsymbol{x}_{m_{t-1}}$. While DiffuseVAE (Pandey et al., 2022) is designed for a VAE with one data modality, we extend it to the multimodal setting and we sample from both the self and cross-modal reconstructions of CMVAE with equal probability:

$$\hat{\boldsymbol{x}}_{m_0} \sim \begin{cases} p_{\theta_m}(\boldsymbol{x}_m|\boldsymbol{z}, \boldsymbol{w}_m), & \begin{subarray}{l} \boldsymbol{z} \sim q_{\phi_{\boldsymbol{z}_m}}(\boldsymbol{z}|\boldsymbol{x}_m), \\ \boldsymbol{w}_m \sim q_{\phi_{\boldsymbol{w}_m}}(\boldsymbol{w}_m|\boldsymbol{x}_m) \end{subarray} & \text{if } b = 0 \quad \textit{self-reconstruction} \\ p_{\theta_m}(\boldsymbol{x}_m|\boldsymbol{z}, \tilde{\boldsymbol{w}}_m), & \begin{subarray}{l} \boldsymbol{z} \sim q_{\phi_{\boldsymbol{z}_n}}(\boldsymbol{z}|\boldsymbol{x}_n), \\ \tilde{\boldsymbol{w}}_m \sim r_m(\boldsymbol{w}_m), \\ n \sim Cat(M, \frac{1}{M}) \end{subarray} & \text{if } b = 1 \quad \textit{cross-reconstruction}, \end{cases} \tag{8}$$

where $b$ is sampled from a Bernoulli distribution with $p = 0.5$. Using the cross-modal reconstructions, which are usually noisier than the self-reconstructions, improves and stabilizes the training, making the diffusion process more robust at test time.

The new objective function is then defined as

$$\mathcal{L}_{\text{D-CMVAE}}(\boldsymbol{X}_0) = \mathcal{L}_{\text{CMVAE}}(\boldsymbol{X}_0) + \frac{1}{M} \sum_{m=1}^{M} \mathbb{E}_{\substack{q_{\phi_{\boldsymbol{z}_m}}(\boldsymbol{z}|\boldsymbol{x}_{m_0}) \\ q_{\phi_{\boldsymbol{w}_m}}(\boldsymbol{w}_m|\boldsymbol{x}_{m_0})}} \mathcal{L}_{\text{DDPM}_m}(\boldsymbol{X}_0, \boldsymbol{z}, \boldsymbol{w}_m) \tag{9}$$

$$\mathcal{L}_{\text{DDPM}_m}(\boldsymbol{X}_0, \boldsymbol{z}, \boldsymbol{w}_m) = \mathbb{E}_{q\left(\boldsymbol{x}_{m_{1:T}}|\boldsymbol{x}_{m_0}\right)} \left[ \frac{p_\psi\left(\boldsymbol{x}_{m_{0:T}} \mid \hat{\boldsymbol{x}}_{m_0}\right)}{q\left(\boldsymbol{x}_{m_{1:T}} \mid \boldsymbol{x}_{m_0}\right)} \right]. \tag{10}$$

As in DiffuseVAE (Pandey et al., 2022), we first train CMVAE using Equation (1) and then we freeze the CMVAE's weights and train the $M$ diffusion processes.

---

**Algorithm 1** Post-hoc selection of optimal latent clusters.

$\bar{H}$ denotes normalized entropy, $|\cdot|$ indicates the number of dimensions of a given latent variable, $\text{assign}_c$ defines the cluster assignment step, and $\text{compute}_\pi$ the prior probability update operation.

---

**Input:** $p_{\boldsymbol{\pi}_K}(\boldsymbol{c}), p(\boldsymbol{z}|\boldsymbol{c}), q_{\phi_{\boldsymbol{z}_1}}(\boldsymbol{z}|\boldsymbol{x}_1) \ldots, q_{\phi_{\boldsymbol{z}_M}}(\boldsymbol{z}|\boldsymbol{x}_M)$ from trained CMVAE with $K > \bar{K}$, data $\boldsymbol{X}^{1:N}$

**Output:** $p_{\boldsymbol{\pi}_{\hat{K}}}$, with $\pi_{\hat{K}}$ s.t. $\sum_{k=1}^{\hat{K}} \mathbb{1}_{\pi_k \neq 0} = \hat{K}$

**for** $k = K$ **to** $2$ **do**

 **for** $\boldsymbol{x}_1^i, \ldots, \boldsymbol{x}_M^i$ **in** $\boldsymbol{x}_1^{1:N}, \ldots, \boldsymbol{x}_M^{1:N}$ **do**

  $\boldsymbol{z}_1^i, \ldots, \boldsymbol{z}_M^i \sim q_{\phi_{\boldsymbol{z}_1}}(\boldsymbol{z}|\boldsymbol{x}_1^i), \ldots, q_{\phi_{\boldsymbol{z}_M}}(\boldsymbol{z}|\boldsymbol{x}_M^i)$

  **for** $m = 1$ **to** $M$ **do**

   $p(\boldsymbol{c}|\boldsymbol{z}_m) = \frac{p_{\boldsymbol{\pi}_k}(\boldsymbol{c})p(\boldsymbol{z}_m|\boldsymbol{c})}{\sum_{\boldsymbol{c}'=1}^k p_{\boldsymbol{\pi}_k}(\boldsymbol{c}')p(\boldsymbol{z}_m|\boldsymbol{c}')}$

  **end for**

  $\boldsymbol{c}_{as}^i = \text{assign}_c(p(\boldsymbol{c}|\boldsymbol{z}_1), \ldots, p(\boldsymbol{c}|\boldsymbol{z}_M))$

  $h_k^i = \frac{1}{M}\sum_{m=1}^M \bar{H}(p(\boldsymbol{c}|\boldsymbol{z}_m)) - \frac{\log p(\boldsymbol{z},\boldsymbol{c})}{|\boldsymbol{z}|}$

 **end for**

 $h_k = \frac{1}{N}\sum_{n=1}^N h_k^n$

 $\boldsymbol{\pi}_{k-1} = \text{compute}_\pi(\boldsymbol{c}_{as}^{1:N}, \boldsymbol{\pi}_k)$

**end for**

$p_{\boldsymbol{\pi}_{\hat{K}}}$ where $\hat{K} = \text{argmin}_k(h_1, \ldots, h_k, \ldots, h_K)$

---

## 4 EXPERIMENTS

We test the performance of CMVAE, in comparison with alternative approaches, on both semi-synthetic and real-world datasets. We first validate our contributions on the PolyMNIST dataset (Sutter et al., 2021), a semi-synthetic five-modality dataset depicting MNIST (LeCun et al., 2010) digits with modality-specific backgrounds, well-established as a benchmark for multimodal VAEs (Sutter et al., 2021; Hwang et al., 2021; Palumbo et al., 2023). In Section 4.1, we compare the generative capabilities of our approach with alternative multimodal VAEs: CMVAE outperforms existing approaches, particularly in unconditional generation. In Section 4.2 we test the clustering capabilities of CMVAE in comparison with alternative unimodal and scalable weakly-supervised approaches. In particular, we validate our proposed post-hoc procedure to determine the optimal set of latent clusters at inference time, hence avoiding the need for a-priori knowledge of the true number of clusters. Finally, in Section 4.3 we validate our contributions in a real-world experiment. We introduce a variation of the CUB Image-Captions dataset (Wah et al., 2011; Shi et al., 2019), which we name the CUB Image-Captions for Clustering (CUBICC) dataset. As the original CUB Image-Captions dataset consists of images of birds paired with matching captions, we group sub-species of birds in the original dataset in eight single species—namely Blackbird, Gull, Jay, Oriole, Tanager, Tern, Warbler, Wren—obtaining a challenging realistic multimodal clustering dataset illustrated in Figure 2b. Details for the datasets are in Appendix D.1.

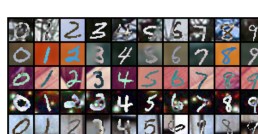

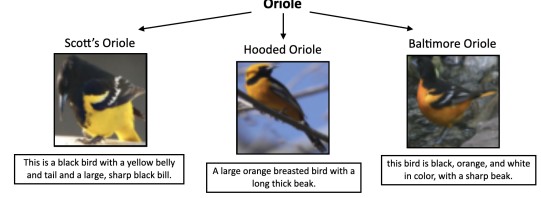

(a) PolyMNIST dataset: Each column is a single data point in this dataset, consisting of five image modalities with matching digit. Each row showcases samples from a given modality.

(b) CUBICC dataset: Example of the variability within a single bird class, Oriole. Here we show the images (modality 1) with their corresponding captions (modality 2).

Figure 2: Illustrative samples for the PolyMNIST (a) and CUBICC (b) datasets.

### 4.1 IMPROVED GENERATION PERFORMANCE OVER EXISTING MULTIMODAL VAEs

In this section, we compare the generative capabilities of our model against the main existing formulations of multimodal VAEs on the PolyMNIST dataset. Successful generative performance of mul-

timodal VAEs requires satisfying two criteria: achieving high semantic coherence across generated modalities (i.e., high generative coherence), as well as having high similarity between generated samples and real samples (i.e. high generative quality). Consistently with previous work (Sutter et al., 2021; Hwang et al., 2021; Palumbo et al., 2023), we compare models in terms of generative quality and generative coherence for both generation with latents sampled from prior distributions (i.e. unconditional generation) and cross-modal generation (i.e., conditional generation), and report our results in Figure 3. In line with prior research (Shi et al., 2019; Daunhawer et al., 2022; Hwang et al., 2021; Palumbo et al., 2023), we adopt the FID score (Heusel et al., 2017) as a proxy for generative quality, and use pre-trained digit classifiers to assess generative coherence. For more details on these metrics, see Appendix D.3. Results show that CMVAE outperforms existing multimodal VAEs, across different hyperparameter values controlling latent space regularization, for both conditional and unconditional generation. While the improvement for conditional generation is moderate, CMVAE outperforms the other methods by a significant margin in unconditional generation, particularly for unconditional coherence. This is an important advancement over state-of-the-art multimodal VAEs, where unconditional generation performance represents a critical weakness.

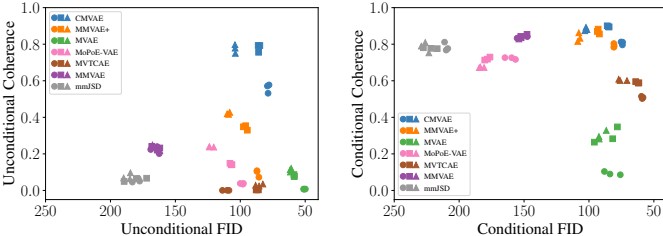

Figure 3: Unconditional (left) and conditional (right) generation performance of CMVAE compared with existing multimodal VAEs on PolyMNIST. Three independent runs for each model, with different symbols denoting different values of the $\beta$ hyperparameter. In both plots, generative coherence is measured on the y-axis (higher is better), while on the x-axis generative quality is assessed via the FID-score (lower is better). Therefore optimal performance is at the top-right corner of each plot.

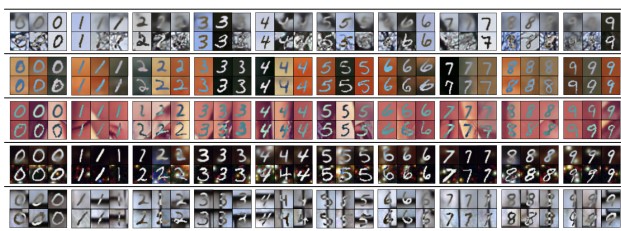

Figure 4: Generative performance for CMVAE and D-CMVAE on the PolyMNIST dataset. Each row showcases generated samples for each modality, with at the top CMVAE generations and corresponding D-CMVAE generations below. We report three generations for all learned clusters, each one corresponding to a different column, across all modalities.

## 4.2 WEAKLY-SUPERVISED CLUSTERING VIA POST-HOC SELECTION OF LATENT CLUSTERS AND SHARP GENERATIONS WITH DDPMS

In this section, we look at clustering results for our method on the PolyMNIST setting and validate the procedure described in Section 3.2 to learn the optimal number of clusters in the data with CMVAE. Specifically, we train multiple instances of CMVAE with $K = 40$ clusters, markedly different from the true number of clusters $\bar{K} = 10$, corresponding to the ten digits. Once an instance of the model is trained, we apply our proposed post-hoc procedure to select the optimal clusters in the latent space, whose effectiveness is showcased in Figure 5. In particular, Figure 5a shows the trend of our proposed entropy-based criterion as the latent clusters are pruned, reaching a minimum when the true latent number of clusters is achieved, consistently across runs. In addition, Figures 5b, 5c, and 5d show how clustering metrics evolve as clusters are pruned: the showcased trends show

that the procedure gradually converges to the solution in which the true clusters are modeled in the latent space, and clustering metrics, therefore, have the best results. These results also confirm that the procedure not only finds the correct number of clusters, but more to the point a minimal set of latent clusters, each one modeling a true cluster in the data. Results in Figure 4 confirm our quantitative analysis from a qualitative standpoint, and in addition show the effectiveness of incorporating DDPMs into our model to increase the quality of generations. Finally in Appendix B, by varying the number of modalities for inference, we show that CMVAE effectively exploits the presence of multiple modalities in computing cluster assignments at test time.

In Table 2 we compare the performance of CMVAE in multimodal clustering with alternative methods. A first natural baseline to report is VaDE (Jiang et al., 2017), a well-established unimodal VAE-based clustering model. However, in order not to restrict ourselves to only variational approaches for unimodal clustering, we include in the comparison the well-known DeepCluster (Caron et al., 2018). In this setting, unimodal clustering approaches fail to achieve good performance (see also Appendix E.2.1): we find they rather model background features, which are prominent pixel-wise in the images, instead of the relevant digit content. As a baseline from the realm of contrastive learning, we adopt CMC (Tian et al., 2020), mainly due to its scalability to numerous modalities, with an instance of $K$-means trained on the learned latent representations. Not surprisingly, this approach achieves comparable yet not superior performance to CMVAE in this setting, as it closely resembles a multi-view setting for which this approach is conceived. However, CMC achieves a comparable performance to our approach on the condition of having a-priori information on the true number of clusters in the data, which is not realistic in real-world scenarios. As shown in Appendix E.2.1, when a-priori information is not available, our method has markedly better performance. Moreover, for more heterogeneous modalities, the gap in performance between the two methods is substantial, as the next section will show. More details about the baselines are reported in Appendix E.2.

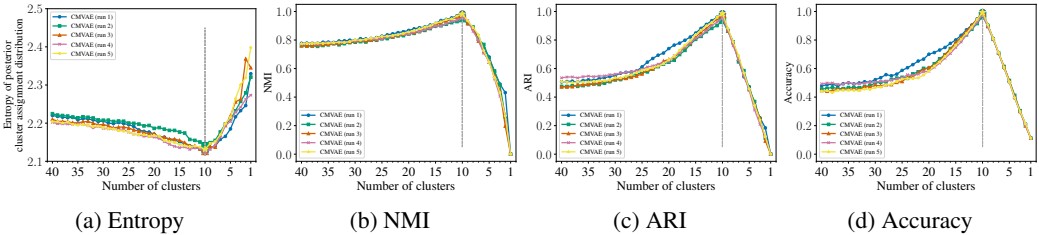

|  | (a) Entropy | (b) NMI | (c) ARI | (d) Accuracy |

Figure 5: Post-hoc optimal latent cluster selection with the procedure in Algorithm 1. (a) Evolution of the entropy term (y-axis) as clusters (x-axis) are pruned. The minimal entropy recovers the true number of clusters. The trends for test NMI (b), ARI (c) and accuracy (d) support the validity of our procedure, showing the procedure selects a minimal set of latent clusters to correctly model the true data clusters. Additional evidence is shown qualitatively in Appendix B.

|  | PolyMNIST | | | CUBICC | | | $\bar{K}$ not given for training |
|---|---|---|---|---|---|---|---|
|  | NMI | ARI | ACC | NMI | ARI | ACC | |
| VaDE | 0.43 (0.04) | 0.36 (0.04) | 0.54 (0.05) | 0.15 (0.01) | 0.08 (0.01) | 0.27 (0.01) | ✗ |
| DeepCluster | 0.12 (0.02) | 0.08 (0.02) | 0.26 (0.04) | 0.19 (0.01) | 0.10 (0.01) | 0.29 (0.01) | ✗ |
| CMC | 0.97 (0.01) | 0.97 (0.01) | 0.99 (0.01) | 0.37 (0.05) | 0.10 (0.03) | 0.31 (0.04) | ✗ |
| CMVAE | 0.97 (0.02) | 0.97 (0.02) | 0.99 (0.01) | 0.67 (0.07) | 0.59 (0.09) | 0.76 (0.07) | ✓ |

Table 2: Quantitative comparison of clustering performance on PolyMNIST and CUBICC datasets.

## 4.3 CUBICC

In this section we test CMVAE on the realistic CUBICC dataset. Note that recent work shows the nature of such a dataset is challenging for most existing multimodal VAEs (Palumbo et al., 2023). Additional challenges arise in modelling data clusters, corresponding to the eight bird species, due to similar appearances (e.g. *Gull* and *Tern*), and high intra-cluster variability as a result of the group-

ing of sub-species (see Figure 2b). In spite of these challenges, our results in Figure 6 and Table 2 show that in this setting CMVAE accurately models the true latent clusters in the data, without prior knowledge of their number and in a completely unsupervised fashion. In particular, results in Appendix B, validate our proposed procedure to find the optimal number of clusters in a real-world setting. A quantitative comparison with alternative approaches in Table 2 validates the effectiveness of CMVAE in this challenging setting, where it outperforms both unimodal and alternative weakly-supervised approaches. In Figure 6 we show the CMVAE and D-CMVAE generations for each learned cluster. As it can be seen, integrating DDPMs into the generative process is highly beneficial to generate sharp images. However, given the scarcity of the training samples and the diversity of bird species, conditioning the diffusion process only on the self-reconstructions produces sub-optimal generative results (see Figure 15 in the Appendix). The integration of cross-modal reconstructions proves to be crucial for obtaining high-quality generations. Finally, in Figure 7, we showcase the performance of D-CMVAE in conditional generation on this dataset, where it generates sharp images, that closely align with text prompts given as input. Notably, D-CMVAE achieves an average FID score of $\approx 28$ on this dataset, a tremendous improvement over generative quality of existing multimodal VAEs in a related setting (see Table 1 in Palumbo et al. (2023)).

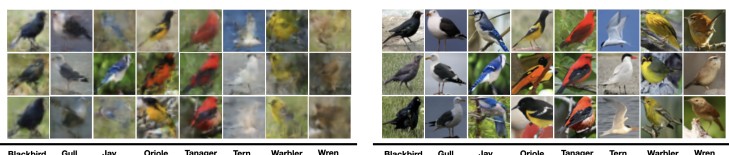

Figure 6: CMVAE (left) and D-CMVAE (right) generations on CUBICC dataset, for each learned cluster. Labels for the clusters are matched by looking at the most frequent label for the samples on the validation set assigned to the given cluster.

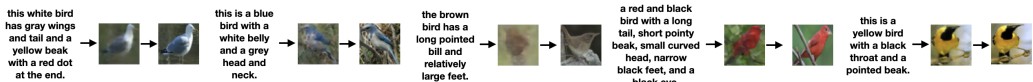

Figure 7: Five instances of conditional caption-to-image generation performance for CMVAE and D-CMVAE: CMVAE and D-CMVAE generations are shown in sequence.

## 5    CONCLUSION

In this work we introduce CMVAE, a novel multimodal VAE, which extends recent advances for multimodal VAEs by enforcing a clustering structure in the latent space. Our experiments show that CMVAE outperforms existing multimodal VAEs in terms of generative performance, in particular showing a remarkable improvement in unconditional generation. Additionally, we show that our proposed model can accurately model data clusters in the latent space, without prior knowledge of the true number of clusters or any other form of supervision. This can be achieved by using a post-hoc procedure to accurately select the optimal number of clusters in the data, without requiring multiple trained instances of the model. Notably, we validate the effectiveness of our approach on the PolyMNIST and the CUBICC datasets. In the latter, introduced as a realistic multimodal clustering setting, CMVAE achieves markedly superior clustering performance compared to both scalable weakly-supervised methods and unimodal approaches. Finally, by incorporating DDPMs into our proposed model we achieve outstanding generative results, previously unseen in multimodal VAEs, significantly boosting the applicability and relevance of our method in real-world scenarios.

**Limitations and future work:** While DDPMs are crucial for high-quality generations, their integration results in a computational overhead both at training and at inference time. Indeed, each modality needs to be modeled by a distinct diffusion process as the different modalities may have varying input dimensions and different features. However, simplification is feasible when the modalities share the same input space, by conditioning a single DDPM on the different modalities. While we proved the effectiveness of our approach on challenging semi-synthetic and realistic datasets, the application to a real-world dataset with a large number of modalities is a main focus for future work.

ACKNOWLEDGEMENTS

EP is supported by a fellowship from the ETH AI Center. EP and DC received funding from the grant #2021-911 of the Strategic Focal Area "Personalized Health and Related Technologies (PHRT)" of the ETH Domain (Swiss Federal Institutes of Technology). LM is supported by the SDSC PhD Fellowship #1-001568-037. SL is supported by the Swiss State Secretariat for Education, Research and Innovation (SERI) under contract number MB22.00047.

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

APPENDIX

TABLE OF CONTENTS

# A   CMVAE OBJECTIVE AS LOWER BOUND ON THE LOG-EVIDENCE

**Lemma A.1.** *The CMVAE objective is a valid lower bound on* $\log p_{\Theta}(\boldsymbol{X})$.

*Proof.* We can prove the proposed CMVAE objective is a valid ELBO using arguments from the proof of Lemma 1 from Palumbo et al. (2023). As described in Section 3.1, we assume the data $\boldsymbol{X} = \boldsymbol{x}_1, \ldots, \boldsymbol{x}_M$ for $M$ modalities is generated according to the following process,

$$p_{\Theta}(\boldsymbol{X}, \boldsymbol{W}, \boldsymbol{z}, \boldsymbol{c}) = p_{\boldsymbol{\pi}}(\boldsymbol{c}) p(\boldsymbol{z}|\boldsymbol{c}) \prod_{m=1}^{M} p_{\theta_m}(\boldsymbol{x}_m|\boldsymbol{w}_m, \boldsymbol{z}) p(\boldsymbol{w}_m),$$

where the latent variables $\boldsymbol{c}, \boldsymbol{z}$ encode cluster assignment and shared information across modalities respectively. Modality-specific latents $\boldsymbol{w}_1, \ldots, \boldsymbol{w}_M$ encode private information for the corresponding modality. Note that shared and modality-specific latents are assumed to be independent. To obtain a tractable lower bound on the log-evidence $\log p_{\Theta}(\boldsymbol{X})$, we start by approximating posterior inference for $\boldsymbol{z}, \boldsymbol{c}$ with an encoder $q_{\Phi_{\boldsymbol{z}}}(\boldsymbol{z}, \boldsymbol{c}|\boldsymbol{X})$. In line with our assumptions for the generative model, we assume the encoder $q_{\Phi_{\boldsymbol{z}}}(\boldsymbol{z}, \boldsymbol{c}|\boldsymbol{X})$ to factorize as $q_{\Phi_{\boldsymbol{z}}}(\boldsymbol{z}, \boldsymbol{c}|\boldsymbol{X}) = q_{\Phi_{\boldsymbol{z}}}(\boldsymbol{z}|\boldsymbol{X}) q(\boldsymbol{c}|\boldsymbol{z}, \boldsymbol{X})$. To achieve scalability in the number of modalities, as in previous works (Shi et al., 2019; Palumbo et al., 2023), we assume a mixture-of-experts encoder for the shared latent code $\boldsymbol{z}$ of the form $q_{\Phi_{\boldsymbol{z}}}(\boldsymbol{z}|\boldsymbol{X}) = \frac{1}{M} \sum_{m=1}^{M} q_{\phi_{\boldsymbol{z}_m}}(\boldsymbol{z}|\boldsymbol{x}_m)$. Hence, we can derive the following ELBO

$$\log p_{\Theta}(\boldsymbol{X}) \geq \mathbb{E}_{\substack{q(\boldsymbol{c}|\boldsymbol{z}, \boldsymbol{X}) \\ q_{\Phi_{\boldsymbol{z}}}(\boldsymbol{z}|\boldsymbol{X})}} \left[ \log \frac{p_{\Theta}(\boldsymbol{X}, \boldsymbol{z}, \boldsymbol{c})}{q_{\Phi_{\boldsymbol{z}}}(\boldsymbol{z}|\boldsymbol{X}) q(\boldsymbol{c}|\boldsymbol{z}, \boldsymbol{X})} \right] \tag{11}$$

$$= \frac{1}{M} \sum_{m=1}^{M} \mathbb{E}_{\substack{q(\boldsymbol{c}|\boldsymbol{z}, \boldsymbol{X}) \\ q_{\phi_{\boldsymbol{z}_m}}(\boldsymbol{z}|\boldsymbol{x}_m)}} \left[ \log \frac{p_{\Theta}(\boldsymbol{X}, \boldsymbol{z}, \boldsymbol{c})}{q_{\Phi_{\boldsymbol{z}}}(\boldsymbol{z}|\boldsymbol{X}) q(\boldsymbol{c}|\boldsymbol{z}, \boldsymbol{X})} \right] \tag{12}$$

$$= \frac{1}{M} \sum_{m=1}^{M} \mathbb{E}_{\substack{q(\boldsymbol{c}|\boldsymbol{z}, \boldsymbol{X}) \\ q_{\phi_{\boldsymbol{z}_m}}(\boldsymbol{z}|\boldsymbol{x}_m)}} \left[ \log \frac{p_{\boldsymbol{\pi}}(\boldsymbol{c}) p(\boldsymbol{z}|\boldsymbol{c}) \prod_{m=1}^{M} p_{\theta_m}(\boldsymbol{x}_m|\boldsymbol{z})}{q_{\Phi_{\boldsymbol{z}}}(\boldsymbol{z}|\boldsymbol{X}) q(\boldsymbol{c}|\boldsymbol{z}, \boldsymbol{X})} \right], \tag{13}$$

which is a sum indexed by the unimodal encoders, where for each term a given unimodal encoder is used for inference. Now, as in (Palumbo et al., 2023), we adopt two different estimators for the self-reconstruction likelihood term $p_{\theta_m}(\boldsymbol{x}_m|\boldsymbol{z})$, i.e. reconstruction of modality used for inference, and cross-reconstruction likelihood terms $p_{\theta_n}(\boldsymbol{x}_n|\boldsymbol{z})$, i.e. reconstruction of modalities *not* used for inference. Specifically, we estimate $\log p_{\theta_m}(\boldsymbol{x}_m|\boldsymbol{z})$ adopting an encoder $q_{\phi_{\boldsymbol{w}_m}}(\boldsymbol{w}_m|\boldsymbol{x}_m)$ for the private latents leading to the lower bound

$$\log p_{\theta_m}(\boldsymbol{x}_m|\boldsymbol{z}) \geq \mathbb{E}_{q_{\phi_{\boldsymbol{w}_m}}(\boldsymbol{w}_m|\boldsymbol{x}_m)} \left[ \log \frac{p_{\theta_m}(\boldsymbol{x}_m|\boldsymbol{w}_m, \boldsymbol{z}) p(\boldsymbol{w}_m)}{q_{\phi_{\boldsymbol{w}_m}}(\boldsymbol{w}_m|\boldsymbol{x}_m)} \right]. \tag{14}$$

In contrast, to estimate cross-reconstruction likelihoods, we adopt the lower-bound

$$\log p_{\theta_n}(\boldsymbol{x}_n|\boldsymbol{z}) = \log \mathbb{E}_{\tilde{\boldsymbol{w}}_n \sim r_n(\boldsymbol{w}_n)} p_{\theta_n}(\boldsymbol{x}_n|\boldsymbol{z}, \tilde{\boldsymbol{w}}_n) \geq \mathbb{E}_{\tilde{\boldsymbol{w}}_n \sim r_n(\boldsymbol{w}_n)} \log p_{\theta_n}(\boldsymbol{x}_n|\boldsymbol{z}, \tilde{\boldsymbol{w}}_n), \tag{15}$$

which uses an auxiliary prior distribution $r_n(\boldsymbol{w}_n)$ specific to each target modality, and is derived by the definition of conditional expectation and Jensen's inequality.

Plugging the derived expressions in (14) and (15) in Equation (13), recovers the CMVAE objective in (1). The fact that (13) is an ELBO and (14) and (15) are lower bounds proves that the CMVAE objective is a valid ELBO. Note a $\beta$ hyperparameter can be plugged in as in (1) to weight latent space regularization. $\qquad\square$

# B   ADDITIONAL CLUSTERING RESULTS FOR CMVAE AND A DEEPER LOOK INTO POST-HOC SELECTION OF OPTIMAL CLUSTERS

In this section, we show additional clustering results for CMVAE on PolyMNIST and CUBICC, with a specific focus on two aspects: the entropy-based post-hoc selection of latent clusters described in Section 3.2 and clustering performance when modalities are missing at test time. In particular, as we

mentioned in Section 2, the efficient handling of missing modalities separates recent advancements in the class of multimodal VAEs from early approaches. In this section we further elaborate on the importance of these features for multimodal clustering models.

**More results for post-hoc selection of optimal latent clusters**  Figure 9 complements the results shown in Figure 5a, by displaying generation results from selected clusters, at different steps of our proposed post-hoc procedure, along with the associated values for entropy of posterior cluster assignments with likelihood penalization. The plot showcases how the minimal value for penalized entropy corresponds to true clusters being modeled in the latent space. In contrast, when $K$ is over-specified certain true clusters are modeled by two or more latent clusters. Finally, when necessary clusters to model the data are pruned, uncertainty in cluster assignments augments, leading to an increase in penalized entropy.

**Post-hoc selection of optimal latent clusters in a realistic setting**  Figure 10 showcases the corresponding quantitative analysis presented in Figure 5 for PolyMNIST on the CUBICC dataset. As such, the results in this section complement those shown in Section 4.3, demonstrating the effectiveness of our proposed entropy-based latent cluster selection procedure also in a realistic scenario. In particular, we train CMVAE on CUBICC with $K = 35$ latent clusters, as assumed in our experiments, and Figure 10a illustrates the trend of our proposed entropy-based criterion as the latent clusters are pruned on the CUBICC dataset. As explained in Appendix D.1, this represents a complex and realistic setting where intra-cluster variability can make it challenging to identify the latent clusters within the data. In spite of this, our post-hoc procedure consistently recovers the true number of clusters, which are accurately modeled in the latent space, with only minor variations across five independent runs. Similar to Figure 5, the trends shown for test NMI, ARI, and accuracy in Figure 10a, Figure 10b and Figure 10c respectively confirm that our procedure not only identifies the correct number of clusters, but also a minimal set of latent clusters necessary to correctly model the data.

**CMVAE clustering performance varying number of modalities present at test time**  In this section, we analyze the clustering performance of CMVAE on PolyMNIST, varying the number of modalities available for inference. As mentioned in Section 2, scalability in the number of modalities is a crucial feature of our proposed approach, setting it apart from alternative methods (e.g. MultiVAE (Xu et al., 2021), XDC (Alwassel et al., 2020)). From the results in Figure 8 and Table 3 we can draw the following important conclusions. First, as more modalities are present for inference, CMVAE clustering performance improves. While this aspect might be easily overlooked, it is crucial that for successful multimodal learning, particularly for the successful performance of multimodal VAEs, adding modalities does not degrade model performance (Shi et al., 2019; Palumbo et al., 2023).

Moreover, we report an additional comparison to validate the importance and effectiveness of clustering in the presence of weak-supervision. Specifically, we train VaDE (Jiang et al., 2017) on the MNIST dataset and compare the results with the ones obtained with CMVAE when a single modality is present at test time. In fact, a single modality of the PolyMNIST dataset essentially presents a slightly more challenging version of the MNIST dataset, where instead of grayscale images, the background is a random crop from a given colored image (Sutter et al., 2021). Notably, CMVAE's clustering performance when tested on a single PolyMNIST modality surpasses that of VaDE trained (and tested) on the grayscale MNIST dataset, where we control for latent space regularization, and VaDE is trained with the correct number of latent clusters, namely 10. This, despite the slightly more challenging nature of a PolyMNIST modality compared to grayscale MNIST. This indicates that weak supervision during training can be beneficial even when testing is done on a single modality.

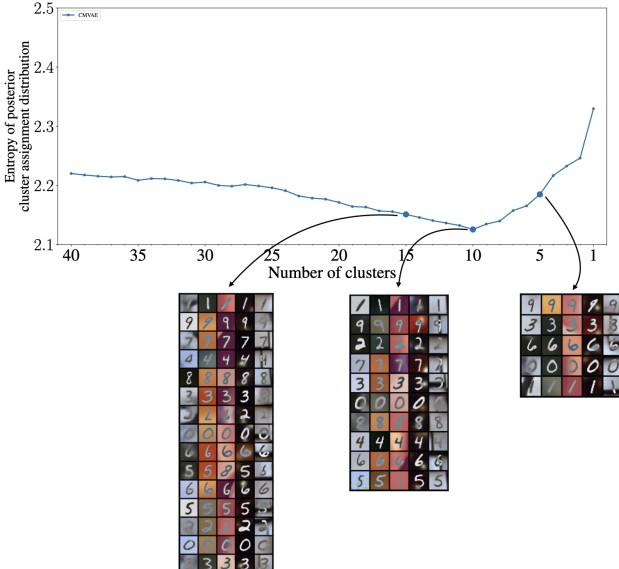

Figure 9: Generated qualitative examples of the clustering capabilities of CMVAE on PolyMNIST dataset. Each column corresponds to the generated samples of one of the five modalities, while each row reports generation results for a different latent cluster.

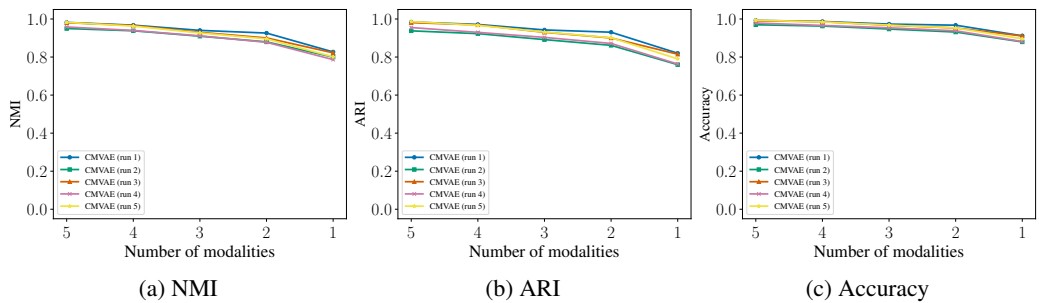

(a) NMI           (b) ARI           (c) Accuracy

Figure 8: CMVAE clustering results in terms of test NMI (a), ARI (b), and accuracy (c) on PolyM-NIST, varying the number of modalities present at test time. Results for five independent runs are shown.

| # modalities | Accuracy | ARI | NMI | | Accuracy | ARI | NMI |
|---|---|---|---|---|---|---|---|
| 5 | 0.985 (0.01) | 0.969 (0.02) | 0.971 (0.02) | VaDE MNIST grayscale | 0.739 (0.07) | 0.619 (0.07) | 0.688 (0.04) |
| 4 | 0.977 (0.01) | 0.952 (0.02) | 0.955 (0.01) | | | | |
| 3 | 0.962 (0.01) | 0.920 (0.02) | 0.924 (0.01) | CMVAE PolyMNIST single modality | 0.900 (0.02) | 0.790 (0.03) | 0.807 (0.02) |
| 2 | 0.949 (0.01) | 0.893 (0.03) | 0.900 (0.02) | | | | |
| 1 | 0.900 (0.02) | 0.790 (0.03) | 0.807 (0.02) | | | | |

Table 3: Left: CMVAE clustering results trained on PolyMNIST, varying the number of modalities present at test time. Right: Comparison of CMVAE trained on PolyMNIST when a single modality is present at test time, and VaDE trained (and tested) on the grayscale MNIST dataset.

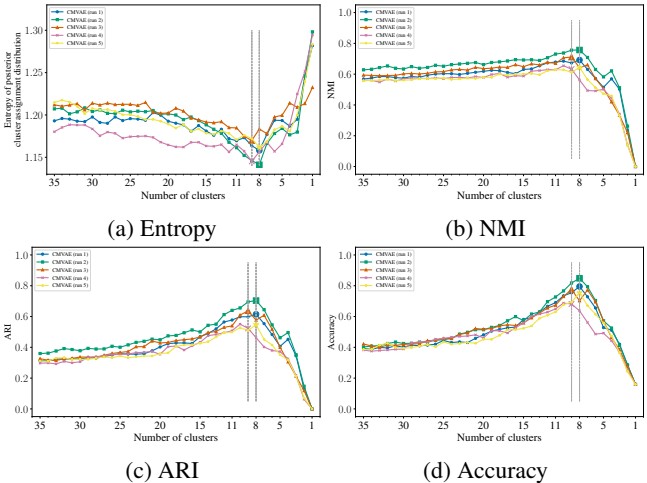

(a) Entropy

(b) NMI

(c) ARI

(d) Accuracy

Figure 10: Post-hoc optimal latent cluster selection with the procedure in Algorithm 1 on the CU-BICC dataset. (a) Evolution of the entropy term (y-axis) as clusters (x-axis) are pruned. The minimal entropy recovers the true number of clusters with only minimal variations. The trends for test NMI (b), ARI (c) and accuracy (c) support the validity of our procedure, showing the procedure selects a minimal set of latent clusters to correctly model the true data clusters.

## C ADDITIONAL QUALITATIVE RESULTS FOR CONDITIONAL AND UNCONDITIONAL GENERATION

In this section, we report qualitative results for CMVAE to complement the quantitative analysis in Figure 3. We do this by providing qualitative examples for both unconditional and conditional generation in Figure 12 and Figure 11, respectively. In particular, here as in Section 4.1 we report generations using CMVAE trained with $\beta = 2.5$, without resorting to the D-CMVAE extension, for a fair comparison with alternative multimodal VAEs tested. Importantly, these results are in line with our quantitative results in Figure 12, in particular the high semantic coherence in unconditional generation, which separates our model from previous approaches is evident. To have a qualitative comparison with the other models reported in Figure 3, we refer to Appendix G.1 in the work of Palumbo et al. (2023). In Figure 13, we show conditional generation results on PolyMNIST when adopting D-CMVAE, which boosts generative quality while at the same time generative coherence is preserved.

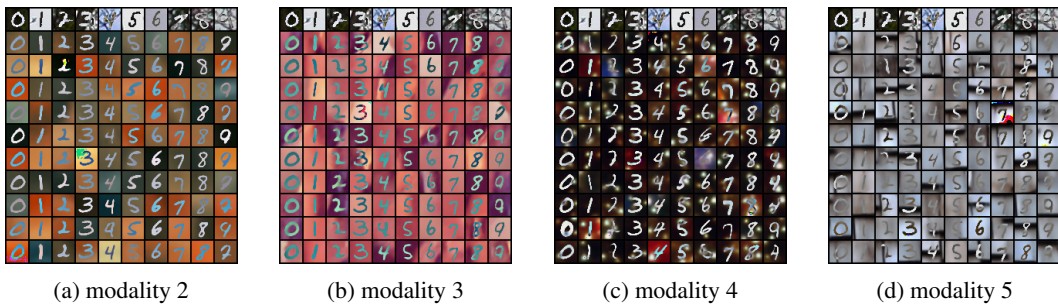

(a) modality 2

(b) modality 3

(c) modality 4

(d) modality 5

Figure 11: Conditional generation from the first modality to the remaining ones for CMVAE trained with $\beta = 2.5$ on the PolyMNIST dataset. In each image, on the top row are starting samples, and below ten instances of conditional generation for the corresponding target modality. Qualitative results complement the analysis in Figure 3.

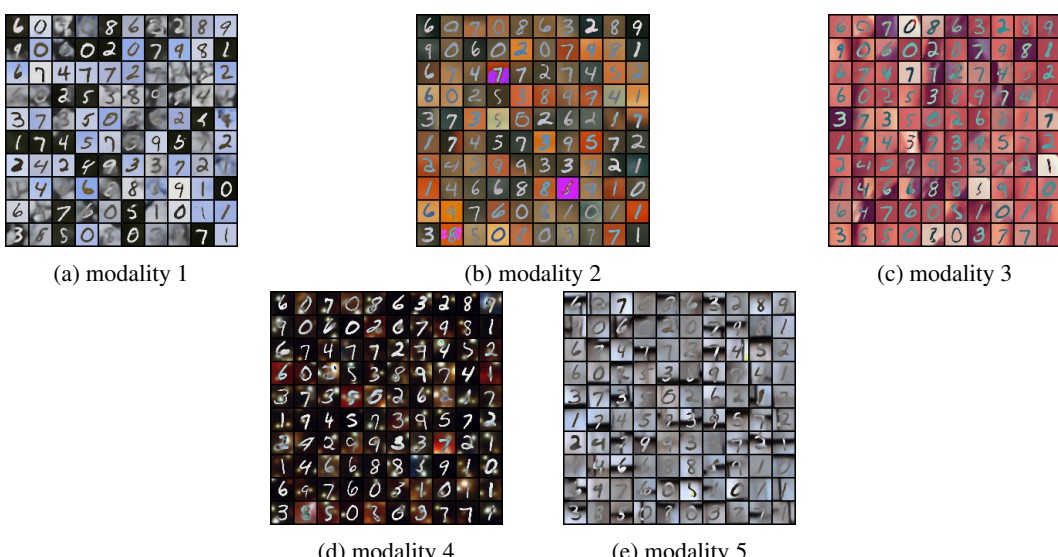

(a) modality 1  (b) modality 2  (c) modality 3

(d) modality 4  (e) modality 5

Figure 12: Uncondtional generation for CMVAE traind with $\beta = 2.5$ on the PolyMNIST dataset. A hundred instances across modalities are shown. Qualitative results complement the analysis in Figure 3.

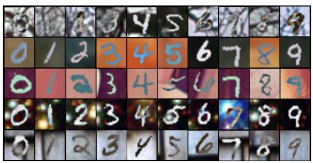

Figure 13: D-CMVAE conditional generation on the PolyMNIST dataset: on the top row are the starting samples from the first modality and on the four remaining rows are the conditional generations in the remaining modalities.

In Figure 14, we show CMVAE unconditional generations on the CUBICC dataset, jointly generating images and captions. As expected, the results show high coherence across the two modalities. Also, as quality of the images in VAE generations is somehow limited, so is quality of text, which could be improved in future work. However, these results favourably compare to related results from recent work (Palumbo et al., 2023). Finally, we perform an ablation on the D-CMVAE model and test the generative performance by conditioning the reverse diffusion process on *only* the self reconstructions of CMVAE. As can be seen in Figure 15, this produces sub-optimal generative results. The integration of the cross-modal reconstructions in the training procedure is crucial to obtain high-quality generations.

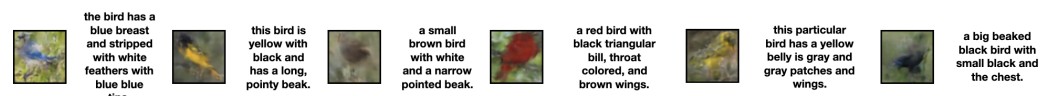

Figure 14: Six instances of CMVAE unconditional generation on the CUBICC dataset, where we jointly generate images and captions. We do not employ D-CMVAE in these results.

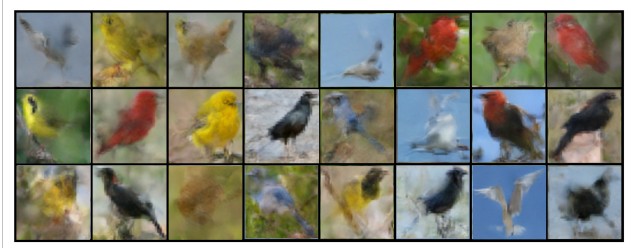

Figure 15: D-CMVAE random generations of CUBICC data by conditioning the reverse diffusion process on *only* the self reconstructions of CMVAE. This results in sub-optimal generative quality.

# D    DETAILS ON DATASETS, METRICS AND IMPLEMENTATION

## D.1    DATASETS

In this section, we provide detailed information about the datasets used in this work. The first dataset we use is the PolyMNIST dataset (Sutter et al., 2021), a semi-synthetic yet challenging dataset consisting of five image modalities: each modality depicts MNIST (LeCun et al., 2010) digits patched on random crops from five distinct background images, one for each modality. Figure 2a showcases some illustrative samples from the dataset. Note that the digit label is the only shared information across modalities, while the background features and the handwriting style differ across modalities in each data point.

As a second experimental setting, we introduce a modified version of the CUB Image-Captions dataset (Wah et al., 2011; Shi et al., 2019). This dataset originally consists of images of birds paired with corresponding descriptive captions. Our version, named the *CUB Image-Captions for Clustering* (CUBICC) dataset, serves as a benchmark for evaluating multimodal clustering methods under more realistic conditions. To create this dataset, we grouped sub-species of birds into a single species category, as illustrated in Figure 16. As a result, the CUBICC dataset consists of eight classes, each representing a different bird species. The grouping of sub-species into a single class introduces significant variability within each class, posing a considerable modeling challenge.

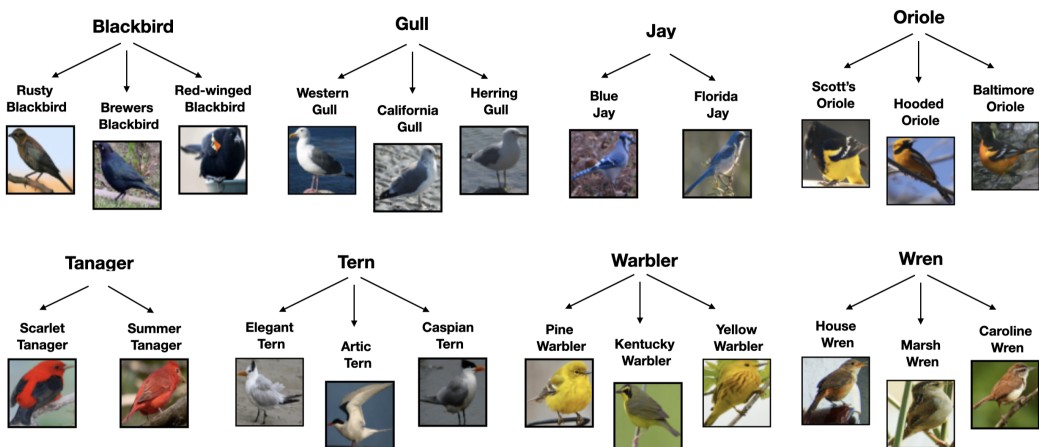

Figure 16: CUBICC dataset structure. Bird sub-species are grouped together in a single category.

## D.2    IMPLEMENTATION DETAILS

To implement all multimodal VAEs (Wu & Goodman, 2018; Shi et al., 2019; Sutter et al., 2020; 2021; Hwang et al., 2021; Palumbo et al., 2023) included in Section 4.1 for comparison in terms of generative performance we follow the same settings as in Palumbo et al. (2023) for training these

models on the PolyMNIST dataset and get best performance. In particular, we use the same ResNet encoder and decoder networks for all compared models. CMVAE is trained for $250$ epochs on this dataset, with $1e^{-3}$ learning rate. As done for MMVAE+, again following the original work of Palumbo et al. (2023), we set both the number of dimensions for the shared and modality-specific latent subspaces of CMVAE to 32 for experiments on PolyMNIST. To train CMVAE on the CU-BICC dataset, we use ResNet and convolutional encoder/decoder networks for the image and text modalities respectively. As in Palumbo et al. (2023), on this dataset we use a 10-sample estimator in our CMVAE ELBO objective and resort to the DReG estimator for gradients (Tucker et al., 2019). Again, we train the model for 250 epochs. We set $64$ dimensions and 32 dimensions for shared and modality-specific latent spaces respectively. For the methods reported as baseline comparisons for clustering tasks, we ensure compatibility with the encoder networks and shared latent space size adopted by CMVAE for a fair comparison. For more details see also Appendix E.2. We follow best practices in their original work for training, but without resorting to any pre-training procedures (for e.g. VaDE (Jiang et al., 2017)), again for a fair comparison with CMVAE. On PolyMNIST and CUBICC, CMVAE is trained with $K = 40$ and $K = 35$ respectively, before the post-hoc procedure described in Section 4.2 is applied. Experimental results across the paper are reported averaging over five independent seeds and we report standard deviations, with the exception of results in Figure 3 and Table 4 where we use three independent seeds. In our clustering experiments for both PolyMNIST and CUBICC datasets we have independent training, validation, and test splits. For PolyMNIST there are 60000 samples for training, 5000 samples for validation, and 5000 samples for testing. For CUBICC, we have 11834 training, 638 validation, and 659 test samples. With CM-VAE, we perform the post-hoc procedure described in Section 4.2 for optimal cluster selection on the validation datasets for both experiments, while clustering results for all models are always reported on the test set. We share the code for our model at `https://github.com/epalu/CMVAE`.

### D.3 METRICS

Consistently with previous work (Sutter et al., 2021; Daunhawer et al., 2022; Hwang et al., 2021; Palumbo et al., 2023), to assess generative quality in our experiments, we resort to the FID score Heusel et al. (2017), which is a well-established metric that has been shown to correlate well with human judgement. Moreover, again consistently with previous work (Shi et al., 2019; Palumbo et al., 2023), we assess generative coherence for our comparisons on the PolyMNIST dataset resorting to the usage of $M$ pre-trained digit classifiers, each one trained on a given modality. In particular, to evaluate $m_1 \rightarrow m_2$ generative coherence, where $m_1, m_2$ are two distinct modalities, we feed conditionally generated samples using $m_1$ as input and $m_2$ as target modality, to the pre-trained digit classifier trained on modality $m_2$. Then we compute the rate with which the given classifier predicts the correct label, i.e., the label from the input samples from modality $m_1$. Finally, to obtain a single metric, we average results for conditional coherence across target and input modalities. To assess generative coherence for unconditional generation, we feed generated samples for each modality to the corresponding pre-trained classifier and compute the rate with which the predicted label matches across *all* modalities.

## E ADDITIONAL AND EXTENSIVE QUANTITATIVE RESULTS

### E.1 QUANTIATIVE RESULTS OF GENERATIVE PERFORMANCE OF MULTIMODAL VAES

Table 4 provides a summary of generative performance, including both average and corresponding standard deviation, derived from three independent runs for the multimodal VAEs illustrated in Figure 3. Both conditional and unconditional generation performance across different values of the regularization hyperparameter $\beta$ are reported.

| $\beta = 1.0$ | Unconditional | | Conditional | |
|---|---|---|---|---|
| | FID | Coherence | FID | Coherence |
| MVAE | 50.65 (0.72) | 0.007 (0.001) | 82.59 (6.22) | 0.093 (0.009) |
| MVTCAE | 110.85 (2.61) | 0.000 (0.000) | 58.98 (0.62) | 0.509 (0.006) |
| mmJSD | 179.76 (2.97) | 0.054 (0.011) | 209.98 (1.26) | 0.785 (0.023) |
| MoPoE-VAE | 98.56 (1.32) | 0.037 (0.002) | 160.29 (4.12) | 0.723 (0.006) |
| MMVAE | 165.17 (3.40) | 0.222 (0.019) | 152.11 (4.11) | 0.837 (0.004) |
| MMVAE+ | 86.64 ( 1.04) | 0.095 (0.020) | 80.75 (0.18) | 0.796 (0.010) |
| CMVAE | 78.52 (0.63) | 0.560 (0.025) | 74.53 (0.64) | 0.806 (0.009) |

| $\beta = 2.5$ | Unconditional | | Conditional | |
|---|---|---|---|---|
| | FID | Coherence | FID | Coherence |
| MVAE | 58.53 (0.12) | 0.080 (0.006) | 85.23 (9.37) | 0.298 (0.044) |
| MVTCAE | 87.07 (0.89) | 0.003 (0.000) | 62.55 (1.30) | 0.591 (0.004) |
| mmJSD | 180.55 (8.67) | 0.060 (0.010) | 222.09 (5.34) | 0.778 (0.003) |
| MoPoE-VAE | 107.11 (0.780) | 0.141 (0.005) | 178.27 (2.01) | 0.720 (0.008) |
| MMVAE | 164.71 (3.17) | 0.232 (0.010) | 150.83 (2.69) | 0.844 (0.010) |
| MMVAE+ | 96.01 (2.10) | 0.344 (0.013) | 92.81 (0.78) | 0.869 (0.013) |
| CMVAE | 85.68 (0.66) | 0.781 (0.021) | 85.12 (0.75) | 0.897 (0.003) |

| $\beta = 5.0$ | Unconditional | | Conditional | |
|---|---|---|---|---|
| | FID | Coherence | FID | Coherence |
| MVAE | 61.25 (0.40) | 0.112 (0.010) | 90.37 (3.20) | 0.301 (0.024) |
| MVTCAE | 85.43 (2.80) | 0.029 (0.001) | 74.61 (3.41) | 0.604 (0.004) |
| mmJSD | 186.49 (2.89) | 0.076 (0.018) | 226.20 (2.91) | 0.784 (0.029) |
| MoPoE-VAE | 122.68 (1.96) | 0.238 (0.001) | 182.99 (1.96) | 0.673 (0.002) |
| MMVAE | 164.29 (2.97) | 0.229 (0.017) | 152.11 (3.18) | 0.839 (0.010) |
| MMVAE+ | 109.08 (1.41) | 0.421 (0.006) | 107.78 (0.88) | 0.836 (0.023) |
| CMVAE | 103.95 (0.16) | 0.775 (0.024) | 102.36 (0.83) | 0.882 (0.010) |

Table 4: Conditional and unconditional generative performance of multimodal VAEs across different values of hyperparameter $\beta$.

## E.2    BASELINE MODELS IN CLUSTERING EXPERIMENTS

In this section, we report further results for the baseline methods we include in our clustering experiments. Specifically, we compare our proposed CMVAE with existing unimodal clustering approaches, namely VaDE (Jiang et al., 2017) and DeepCluster (Caron et al., 2018), as well as with weakly-supervised multimodal approaches, namely XDC (Alwassel et al., 2020) and CMC (Tian et al., 2020). The reported results in the main text correspond to giving as input the true a-priori number of clusters to the model. In particular for the unimodal approaches, we select the best performance across all modalities. It is important to note that all these methods require a pre-specified value for the number of clusters $K$, unlike CMVAE. On PolyMNIST, we test these models using $K = 100$, $K = 40$, and $K = 10$. The rationale behind this choice is as follows: testing with $K = 40$ allows for a fair comparison with CMVAE, which is trained with $K = 40$ in our PolyM-NIST experiments, providing a fair and realistic scenario where prior information on the true number of clusters is unavailable. Furthermore, we evaluate these methods with $K = 100$ to give them a large modeling capacity. Finally, we explore a facilitated setting where the true number of clusters $K = 10$ is given for training. Similarly, in the CUBICC dataset, we evaluate all corresponding baseline methods with $K = 100$, $K = 35$, and $K = 8$ clusters, following the same rationale as in the PolyMNIST experiment.

### E.2.1    POLYMNIST EXPERIMENT

**Unimodal approaches**    Here we present the results for the unimodal clustering approaches on the PolyMNIST dataset. We evaluate VaDE using different values of the regularization hyperparameter $\beta$ in the VAE objective, specifically 1.0, 2.5, and 5.0, selecting the value corresponding to best

performance for our comparisons in the main text. The results for these approaches trained on each modality are reported in Table 5. Values reported in the main text are underlined.

PolyMNIST: Unimodal clustering

| | | | m0 | m1 | m2 | m3 | m4 |
|---|---|---|---|---|---|---|---|
| VaDE | $\beta$=1.0 | NMI | 0.031 (0.01) | 0.123 (0.05) | 0.013 (0.01) | 0.391 (0.04) | 0.004 (0.00) |
| | | ARI | 0.011 (0.00) | 0.074 (0.03) | 0.005 (0.01) | 0.313 (0.04) | 0.000 (0.00) |
| | | Acc | 0.147 (0.01) | 0.231 (0.04) | 0.130 (0.01) | 0.500 (0.05) | 0.120 (0.00) |
| | $\beta$=2.5 | NMI | 0.025 (0.01) | 0.154 (0.05) | 0.008 (0.01) | 0.431 (0.04) | 0.007 (0.00) |
| | | ARI | 0.010 (0.01) | 0.094 (0.03) | 0.002 (0.00) | 0.364 (0.04) | 0.002 (0.00) |
| | | Acc | 0.141 (0.01) | 0.262 (0.04) | 0.124 (0.01) | 0.544 (0.05) | 0.127 (0.00) |
| | $\beta$=5.0 | NMI | 0.031 (0.02) | 0.101 (0.06) | 0.035 (0.04) | 0.415 (0.09) | 0.004 (0.00) |
| | | ARI | 0.014 (0.01) | 0.054 (0.03) | 0.018 (0.02) | 0.355 (0.10) | 0.000 (0.00) |
| | | Acc | 0.145 (0.02) | 0.208 (0.04) | 0.147 (0.03) | 0.538 (0.10) | 0.120 (0.00) |
| DeepCluster | $K$=10 | NMI | 0.023 (0.01) | 0.038 (0.02) | 0.023 (0.01) | 0.122 (0.02) | 0.006 (0.00) |
| | | ARI | 0.009 (0.01) | 0.015 (0.01) | 0.010 (0.01) | 0.079 (0.02) | 0.002 (0.00) |
| | | Acc | 0.140 (0.01) | 0.125 (0.06) | 0.143 (0.013) | 0.258 (0.04) | 0.124 (0.00) |
| | $K$=35 | NMI | 0.089 (0.01) | 0.132 (0.03) | 0.076 (0.02) | 0.141 (0.02) | 0.013 (0.00) |
| | | ARI | 0.024 (0.00) | 0.047 (0.01) | 0.023 (0.01) | 0.051 (0.01) | 0.001 (0.00) |
| | | Acc | 0.096 (0.00) | 0.138 (0.01) | 0.100 (0.02) | 0.145 (0.01) | 0.060 (0.00) |
| | $K$=100 | NMI | 0.163 (0.02) | 0.210 (0.03) | 0.105 (0.02) | 0.180 (0.02) | 0.033 (0.00) |
| | | ARI | 0.026(0.00) | 0.039 (0.01) | 0.014 (0.09) | 0.031 (0.01) | 0.001 (0.00) |
| | | Acc | 0.069 (0.01) | 0.088 (0.01) | 0.054 (0.01) | 0.081 (0.01) | 0.031 (0.00) |

Table 5: Baseline results for unimodal clustering on PolyMNIST dataset. Including ablation of $\beta$ for VaDE and number of clusters for DeepCluster. Mean and standard deviation of five independent runs are reported. The underlined results correspond to those reported in Table 2 in the main text.

**Multimodal approaches** In addition, we present results on the multimodal benchmarks. XDC is designed for two modalities and it does not directly scale to a larger number of modalities, hence, we do not include it in the main text, but we show it here for reference. To adapt XDC for PolyM-NIST, we select uniformly at random which other modality to extract the pseudo-labels from at each iteration. Nevertheless, its low performance highlights that the method is not properly scalable. The second baseline, CMC, is a contrastive-based approach for multi-view representation learning. To evaluate its clustering capabilities, we train an instance of $K$-means on the embedding space. As discussed in Section 4.2, the high performance of CMC on PolyMNIST is on par with our proposed CMVAE. However, this performance is obtained only when the model has a-priori knowledge of the number of clusters, which is not realistic in practice. Particularly, in Table 6 we can see how without a-priori information, i.e. K being 40 as in CMVAE, CMC performance decreases significantly and is no-longer on par with CMVAE. For this analysis, we explore values of $K$ 10, 20, 35, 40, and 100 to illustrate how such high performance is only attainable with an oracle-informed a-priori $K$. For both CMC and XDC, the final metrics are reported in Table 6 after averaging the embedding space generated by each modality and clustering using $K$-means on the average space. The results used in the main text are underlined in Table 6 .

PolyMNIST: Multimodal clustering

| | | | |
|---|---|---|---|
| XDC | $K$=10 | NMI | 0.020 (0.01) |
| | | ARI | 0.009 (0.01) |
| | | Acc | 0.145 (0.01) |
| | $K$=35 | NMI | 0.118 (0.13) |
| | | ARI | 0.053 (0.05) |
| | | Acc | 0.102 (0.05) |
| | $K$=40 | NMI | 0.122 (0.07) |
| | | ARI | 0.038 (0.03) |
| | | Acc | 0.100 (0.02) |
| | $K$=100 | NMI | 0.360 (0.08) |
| | | ARI | 0.082 (0.02) |
| | | Acc | 0.101 (0.02) |
| CMC | $K$=10 | NMI | 0.970 (0.01) |
| | | ARI | 0.969 (0.02) |
| | | Acc | 0.986 (0.01) |
| | $K$=20 | NMI | 0.876 (0.01) |
| | | ARI | 0.706 (0.02) |
| | | Acc | 0.617 (0.02) |
| | $K$=35 | NMI | 0.800 (0.00) |
| | | ARI | 0.480 (0.01) |
| | | Acc | 0.407 (0.01) |
| | $K$=40 | NMI | 0.784 (0.00) |
| | | ARI | 0.448 (0.01) |
| | | Acc | 0.380 (0.01) |
| | $K$=100 | NMI | 0.685 (0.00) |
| | | ARI | 0.261 (0.11) |
| | | Acc | 0.219 (0.09) |

Table 6: Baseline results for multimodal clustering on the PolyMNIST dataset, including ablation of the number of clusters $K$. Underlined results are reported in the main text.

### E.2.2 CUBICC EXPERIMENT

**Unimodal and multimodal approaches** In this section, we present results for the clustering approaches on the CUBICC dataset. As in the PolyMNIST dataset, we examine VaDE's performance using different values of the $\beta$ hyperparameter. We also test DeepCluster with varying $K$. Results are reported in Table 7. Note that only the image modality is reported for DeepCluster, as it is a method for image data. The multimodal results are displayed in Table 8. Unlike the PolyMNIST dataset, CUBICC involves two modalities, aligning with XDC's original design, which is reflected in the performance improvement observed. In both tables, the results reported in the main manuscript are underlined.

**Latent space ablation** Previous results in this experimental setting used a 64-dimensional latent space. Here we increase the latent space size to 96 dimensions and show in Table 9 that this increase in latent space capacity does not result in significant changes in performance for the reported methods.

CUBICC: Unimodal clustering

| | | | Image | Text |
|---|---|---|---|---|
| VaDE | $\beta$=1.0 | NMI | 0.079 (0.02) | 0.021 (0.00) |
| | | ARI | 0.041 (0.01) | 0.002 (0.00) |
| | | Acc | 0.243 (0.03) | 0.180 (0.00) |
| | $\beta$=2.5 | NMI | 0.112 (0.03) | 0.020 (0.00) |
| | | ARI | 0.063 (0.02) | 0.000 (0.00) |
| | | Acc | 0.259 (0.02) | 0.174 (0.00) |
| | $\beta$=5.0 | NMI | 0.155 (0.01) | 0.019 (0.00) |
| | | ARI | 0.084 (0.01) | 0.000 (0.00) |
| | | Acc | 0.269 (0.01) | 0.171 (0.00) |
| DeepCluster | $K$=8 | NMI | 0.191 (0.01) | - |
| | | ARI | 0.100 (0.01) | - |
| | | Acc | 0.294 (0.01) | - |
| | $K$=35 | NMI | 0.247 (0.00) | - |
| | | ARI | 0.058 (0.00) | - |
| | | Acc | 0.146 (0.00) | - |
| | $K$=100 | NMI | 0.310 (0.01) | - |
| | | ARI | 0.035 (0.00) | - |
| | | Acc | 0.099 (0.01) | - |

Table 7: Baseline results for unimodal clustering on test images from CUBICC dataset, including ablation of $\beta$ for VaDE and number of clusters $K$ for DeepCluster. Underlined results are the ones reported in Section 4.2 in the main text, namely the ones corresponding to the best modality and the best $\beta$ for VaDE and to the true number of clusters for DeepCluster.

CUBICC: Multimodal clustering

| | | | |
|---|---|---|---|
| XDC | $K$=8 | NMI | 0.218 (0.06) |
| | | ARI | 0.109 (0.02) |
| | | Acc | 0.281 (0.05) |
| | $K$=35 | NMI | 0.262 (0.02) |
| | | ARI | 0.116 (0.10) |
| | | Acc | 0.224 (0.10) |
| | $K$=100 | NMI | 0.323 (0.00) |
| | | ARI | 0.035 (0.00) |
| | | Acc | 0.091 (0.01) |
| CMC | $K$=8 | NMI | 0.374 (0.05) |
| | | ARI | 0.097 (0.03) |
| | | Acc | 0.310 (0.04) |
| | $K$=35 | NMI | 0.586 (0.02) |
| | | ARI | 0.313 (0.07) |
| | | Acc | 0.385 (0.09) |
| | $K$=100 | NMI | 0.556 (0.01) |
| | | ARI | 0.180 (0.02) |
| | | Acc | 0.211 (0.02) |

Table 8: Baseline results for multimodal clustering on CUBICC dataset. Including ablation of number of clusters. Underlined results are the ones reported in Section 4.2 in the main text.

CUBICC: 96-dimensional latent space

| | | |
|---|---|---|
| Vade $\beta$=5.0 | NMI | 0.178 (0.01) |
| | ARI | 0.103 (0.01) |
| | Acc | 0.297 (0.00) |
| DeepCluster $K$=8 | NMI | 0.195 (0.02) |
| | ARI | 0.106 (0.02) |
| | Acc | 0.304 (0.02) |
| XDC $K$=8 | NMI | 0.246 (0.04) |
| | ARI | 0.142 (0.03) |
| | Acc | 0.339 (0.02) |
| CMC $K$=8 | NMI | 0.313 (0.05) |
| | ARI | 0.124 (0.12) |
| | Acc | 0.268 (0.04) |

Table 9: Results of clustering baselines on a latent space of 96 dimensions for the CUBICC dataset.

