# OpenReview forum: "Deep Generative Clustering with Multimodal Diffusion Variational Autoencoders"
_ICLR.cc/2024/Conference — ICLR 2024 poster_

### Official Review · Reviewer_GVdd · 2023-10-26

**Soundness:** 3 good
**Presentation:** 2 fair
**Contribution:** 3 good
**Rating:** 8
**Confidence:** 3

**Summary:**

This paper introduces a novel multimodal Variational Autoencoder (VAE) model that combines weakly-supervised learning and clustering for multiple heterogeneous modalities. The key contributions of the paper are as follows:

**Multimodal VAE Model:** The paper presents a new multimodal VAE model that extends the latent space to simultaneously learn data clusters while leveraging shared information across different data modalities.

**Improved Generative Performance:** Experimental results demonstrate that the proposed model outperforms existing multimodal VAEs in terms of generative performance, particularly for unconditional generation tasks.

**Automatic Cluster Selection:** The paper introduces a post-hoc procedure to automatically determine the number of true clusters, addressing critical limitations of previous clustering frameworks. This enhances the model's ability to discover meaningful clusters in weakly-supervised settings.

**Comparison to Alternative Clustering Approaches:** The proposed method is compared favorably to alternative clustering approaches in weakly-supervised settings, highlighting its effectiveness in clustering and learning representations from multimodal data.

Integration of Diffusion Models: The paper incorporates recent advancements in diffusion models into the proposed method to further enhance generative quality, particularly for real-world images. This integration helps improve the realism of generated images.

**Strengths:**

I appreciate the authors' great effort in providing various experimental studies in the main part as well as the appendix.

* The paper is well-organized.
* The results show clear improvement over the baseline methods, especially for unconditional generation and realistic data clustering.
* The idea is incrementally inspired from recent MVAE methods as well as DiffuseVAE; But, it is sound and interesting.
* The automatic cluster selection on test data is interesting and novel, showing advantage to the prior work.

**Weaknesses:**

The proposed method needs better explanations.

* What are the learned modules in CMVAE? Are all $q(z | X)$, $q(w_i | x_i)$, $p(w_i)$, $p(c)$, $p(z|c)$ learned during the training?
* In figure 1, I cannot understand why there exist $z_1$, ..., $z_M$.
* I cannot understand the cross-reconstruction formulation in Eq (8). I was thinking $p_{\theta_m}$ can be only called with $w_m$

**Questions:**

The questions were asked in the weaknesses part.

BTW, what are the cluster in CUBICC dataset? species or the subspecies?

---

> ### Author Response · Authors · 2023-11-18
> **Reply to Reviewer GVdd**
>
> We thank the Reviewer for the positive feedback on the relevance, soundness, and organization of our work.
>
> In the following we address the questions from the Reviewer.
>
> > What are the learned modules in CMVAE? Are all $q(z| X), q(w_i| x_i), p(w_i), p(c), p(z|c)$ learned during the training?
>
> The modules $q_{\Phi_z}(z | X), q_{\phi_{w_1}}(w_1| x_1), \dots, q_{\phi_{w_M}}(w_M| x_M), p_{\pi}(c), p(z|c), p_{\theta_1}(x_1|w_1, z), \dots, p_{\theta_M}(x_M|w_M, z)$ are learned during training. The prior distributions for modality-specific features $p(w_1), \dots, p(w_M)$ are set to zero-mean unit-variance distributions.  Following the recent work by Palumbo et al.[1], the auxialiary distributions for modality-specific features used at training time to facilitate the estimation of cross-modal reconstructions are zero-mean distributions with parameterized variance that is learned at training time. We denote these distributions by $r_1(w_1), \dots, r_M(w_M)$ in the updated version of the manuscript, uniforming our notation with the work from Palumbo et al. [1].
>
> > In figure 1, I cannot understand why there exist $z_1, \dots, z_M$.
>
> Note that we model the joint encoder for $z$ as a mixture-of-experts $q_{\Phi_z}(z| X) = \frac{1}{M} \sum_{m=1}^M q_{\phi_{z_m}}(z| x_m)$, and hence computing our ELBO requires sampling $M$ latent codes for $z$, each one sampled by the corresponding unimodal encoder $q_{\phi_{z_m}}(z| x_m)$. That is why in Figure 1 $z_1, \dots, z_M$ are present.
>
> > I cannot understand the cross-reconstruction formulation in Eq (8). I was thinking $p_{\theta_m}$ can be only called with $w_m$.
>
> As mentioned above, as in previous work from Palumbo et al. [1] we sample modality-specific features from auxiliary distributions for cross-reconstruction. Looking at the updated version of the manuscript, where we uniform the notation with Palumbo et al. [1], the cross-reconstruction formula in Eq. (8) is $p_{\theta_m}(x_m|z, \tilde{w}_n)$ , where $\tilde{w}_n$ is sampled from the auxiliary distribution $r_n(w_n)$ for the target modality $n$.
>
> > BTW, what are the cluster in CUBICC dataset? species or the subspecies?
>
> The clusters in the CUBICC dataset are the eight species: Blackbird, Gull, Jay, Oriole, Tanager, Tern, Warbler, Wren. We further clarify this point in the updated version of the manuscript.
>
> We hope with this rebuttal we addressed all the questions from the Reviewer.
>
> **References:**
>
> [1] Palumbo et al. MMVAE+: Enhancing the generative quality of multimodal VAEs without compromises. In ICLR, 2023.

---

> > ### Comment · Reviewer_GVdd · 2023-11-22
> > **Response to Rebuttal**
> >
> > I appreciate the authors' responses to my questions.
> >
> > Just a question about sampling from $q_{\Phi_z(z | X)}$. Is sampling from this mixture-of-experts distribution performed as follows? You first randomly select one of the experts and then sample from its corresponding Gaussian distribution?

---

> > > ### Author Response · Authors · 2023-11-22
> > > **Response to follow-up question**
> > >
> > > Thank you for your feedback and for your question.
> > >
> > > Generally, if one is to sample from the joint mixture-of-experts encoder $q_{\Phi_z}(z|X)$, your understanding is correct that one would first sample an expert uniformly at random and then sample from the corresponding distribution. However, note that to compute our ELBO a latent code for $z$ is sampled from each corresponding expert (unimodal encoder), hence no sampling of the experts is involved.
> > >
> > > We hope this comment addresses your question.

---

### Official Review · Reviewer_XPst · 2023-10-29

**Soundness:** 3 good
**Presentation:** 3 good
**Contribution:** 2 fair
**Rating:** 6
**Confidence:** 3

**Summary:**

The authors present a neat extension of typical VAEs to the multi-modal settings. It results in a framework that can be used for weakly-supervised clustering. By further combining with Denoising Diffusion Probabilistic Models, it can generative better multi-modal

**Strengths:**

- The presentation is clear and well-structured.
- The authors report solid experimental results, both qualitatively and quantitatively on the selected two datasets.
- The method is technically sound, and several prior works on VAEs are neatly integrated together.

**Weaknesses:**

1. I'm concerned about the technical novelty. The authors extend typical VAEs to multi-modal VAEs for two tasks. The first is the application of weakly-supervised clustering, and the second is to combine multi-modal VAEs with DDPM to improve its generative quality. However, both applications of VAEs are not novel and can be viewed as an extension from VAEs to multi-modal VAEs. Though some new tricks are proposed in the form of multi-modal VAEs like Eq.8, it is not properly discussed and evaluated with sufficient ablation studies.

2. The experiments are only conducted on one realistic multi-modal dataset that only contains two modalities. The authors mention the state-of-the-art weakly supervised methods are not scalable to numerous modalities while the proposed method is. The prior work is either on image-text [1] datasets or video-audio [2] [3] datasets. In Table 2, on PolyMNIST the proposed method is not improved over non-VAE SOTA either.
Therefore, I think the authors overclaim the contribution of the scalability of numerous modalities. Especially for weakly-supervised clustering, in my opinion, a realistic multi-modal dataset is essential to claim the contribution of scalability to more modalities.


[1] Zhou et al., End-to-End Adversarial-Attention Network for Multi-Modal Clustering.

[2] Chen et al., Multimodal Clustering Networks for Self-supervised Learning from Unlabeled Videos.

[3] Alwassel et al., Self-Supervised Learning by Cross-Modal Audio-Video Clustering.

**Questions:**

1. Since the contribution of this work is the combination of several existing approaches based on VAEs. It is important for the authors to focus on to further justify the contribution of extension VAEs to multi-modal VAEs in these tasks are not trivial and require careful design. I suggest the authors conduct ablation studies of such designs e.g. Eq.8.

2. To claim the scalability of the method, the authors may consider more experiments on realistic datasets with more than two modalities.

### After rebuttal
Thanks for the response and addressing my concerns. I raised my score to borderline accept. I suggest the authors consider moving some content in the Appendix (e.g. Eq.8 ablation study) to the main manuscript.

---

> ### Author Response · Authors · 2023-11-18
> **Reply to Reviewer XPst (Part 1)**
>
> We thank the Reviewer for the positive feedback on the clarity and structure of the paper, as well as on the soundness of our work and the presented experimental evidence.
>
> In the following, we address the points raised by the Reviewer concerning novelty and scalability of the proposed approach.
>
> > I’m concerned about the technical novelty. The authors extend typical VAEs to multi-modal VAEs for two tasks. The first is the application of weakly-supervised clustering, and the second is to combine multi-modal VAEs with DDPM to improve its generative quality. However, both applications of VAEs are not novel and can be viewed as an extension from VAEs to multi-modal VAEs. Though some new tricks are proposed in the form of multi-modal VAEs like Eq.8, it is not properly discussed and evaluated with sufficient ablation studies.
>
> As we argue in our paper, we position this work at the gist of two existing research directions: multimodal VAEs and (unimodal) VAE-based approaches for clustering. As highlighted by the other reviewers, combining these research areas in a holistic, versatile approach to improve over state-of-the-art models is novel and proved relevant and effective with solid empirical performance advancements in both settings.
>
> We propose a novel multimodal VAE model incorporating clustering in the latent space, which advances the state-of-the-art of multimodal VAEs (see Section 4.1). Note that this improvement in generative capabilities is only due to modeling advancements since, for these comparisons, we do not use DDPMs for generation. While multimodal VAEs and their cross-modal generation capabilities have recently gained significant attention, improving unconditional generation has been an overlooked topic [1][2]. In our work, we tackle this aspect and achieve a marked improvement in unconditional generation performance, therefore contributing to the field of multimodal VAEs.
>
> In addition, we propose a novel approach to perform clustering on datasets consisting of numerous modalities, and we outperform existing scalable approaches. (Note: for evidence for this claim, please refer also to our answer to the point below). We also incorporate a novel post-hoc procedure to automatically select the optimal latent clusters at test time, avoiding the need to specify the correct number of clusters a-priori, thus greatly enhancing the applicability of the proposed method in real-world settings.
>
> Finally, while DDPMs have been incorporated into the VAE framework, we are, to the best of our knowledge, the first ones to investigate their integration in multimodal VAEs. In particular, and specifically to your question
>
>
> > I suggest the authors conduct ablation studies of such designs e.g. Eq.8.
>
>
> we provide several insights on the integration of DDPMs in mixture-based multimodal VAEs. In particular, we performed ablation experiments to empirically validate Eq.8 in the Appendix (see Fig.15). We showed that conditioning the DDPM on the self-reconstructions alone is insufficient to generate high-quality images. Specifically, cross-reconstructions are usually noisier than self-reconstructions as modality-specific information cannot be inferred from the input sample. Therefore, by alternating between self and cross-reconstructions, the DDPM is more robust against noisy VAE generations and cross-generations, greatly improving generative performances (see Fig.6). The integration of DDPMs in the multimodal VAE framework proves to have a massive impact on generative capabilities for both unconditional and conditional generation and is an important step towards the applicability of multimodal VAEs in complex real-world settings.
>
> **[Note: this reply continues in a separate comment.]**

---

> ### Author Response · Authors · 2023-11-18
> **Reply to Reviewer XPst (Part 2)**
>
> > The experiments are only conducted on one realistic multi-modal dataset that only contains two modalities. The authors mention the state-of-the-art weakly supervised methods are not scalable to numerous modalities while the proposed method is. The prior work is either on image-text [1] datasets or video-audio [2] [3] datasets. In Table 2, on PolyMNIST the proposed method is not improved over non-VAE SOTA either. Therefore, I think the authors overclaim the contribution of the scalability of numerous modalities. Especially for weakly-supervised clustering, in my opinion, a realistic multi-modal dataset is essential to claim the contribution of scalability to more modalities.
>
> > [1] Zhou et al., End-to-End Adversarial-Attention Network for Multi-Modal Clustering.
>
> > [2] Chen et al., Multimodal Clustering Networks for Self-supervised Learning from Unlabeled Videos.
>
> > [3] Alwassel et al., Self-Supervised Learning by Cross-Modal Audio-Video Clustering.
>
> Note that from our results on PolyMNIST, our approach *does* represent a relevant improvement over non-VAE SOTA. In fact, while the performance of CMC and PolyMNIST matches in Table 2, CMC was given the true number of clusters (ten) as a-priori information. In the realistic setting in which the true number of clusters in unknown a-priori (see Table below and Appendix E of our paper), our method significantly surpasses other non-VAE SOTA. This validates the importance of our novel procedure to automatically select optimal clusters in the data. In the updated manuscript, we highlight this point when commenting the results in Table 2 and elaborate further in Appendix E.
>
> As for the claim that our model is a scalable approach in the number of modalities, this rather refers to the fact that our approach is a multimodal VAE model that can handle numerous modalities efficiently for training and inference, in comparison e.g. to initial approaches in the class of multimodal VAEs [3][4] (see first paragraph of our Related Work section). Note this characterization is well-established in the multimodal VAEs literature [2][5][6], and following this characterization our model falls into the family of scalable multimodal VAEs. The fact that our model can handle a large number of modalities present for training and inference already separates our approach from other weakly-supervised clustering models [7][8], that cannot be trivially extended to handle more than two modalities for training, or to perform inference if only a subset of modalities is present at test time.
>
> When testing our model on the PolyMNIST dataset we show it can handle numerous modalities and exploit information from these modalities to obtain high clustering performance. While a synthetic dataset, PolyMNIST proves to be already challenging for existing approaches, and to back up this claim we also show a comparison with the recently proposed XDC [7] in this setting (see Appendix E in the manuscript and Table below), which fails to achieve satisfactory performace. Originally, XDC is proposed for two modalities, where the pseudo-labels for one modality's classification are derived from clustering assignments from the embedding space of the other modality, and does not trivially extend to more than two modalities. Hence, to adapt XDC to scale to more than two modalities, we select uniformly at random which other modality to extract the pseudo-labels from at each iteration. Its limited performance on PolyMNIST shows that it struggles to capture the relevant information from more than two modalities and confirms the relevance of the results obtained with CMVAE. Finally, in Table 3 in the Appendix we show CMVAE's test-time clustering performance on PolyMNIST improves when more modalities are present for inference, which indicates our model can effectively exploit additional modalities for better clustering.
>
> |              | NMI         | ARI         | ACC         | K not needed for training |
> |--------------|-------------|-------------|-------------|---------------------------|
> | XDC          | 0.02 (0.01) | 0.01 (0.01) | 0.15 (0.01) | no                        |
> | CMC          | 0.97 (0.02) | 0.97 (0.02) | 0.99 (0.01) | no                        |
> | CMC          | 0.78 (0.00) | 0.45 (0.01) | 0.35 (0.01) | yes                       |
> | CMVAE (ours) | 0.97 (0.02) | 0.97 (0.02) | 0.99 (0.01) | yes                       |
>
>
> **[Note: this reply continues in a separate comment.]**

---

> ### Author Response · Authors · 2023-11-18
> **Reply to Reviewer XPst (Part 3)**
>
> While we make these considerations, we also appreciate the comment and suggestion from the Reviewer and agree on the value of a realistic multimodal dataset with a large number of modalities to benchmark clustering approaches. Therefore in the updated version of the manuscript (precisely under Limitations and Future work) we clarify that, applying our approach to a real-world multimodal dataset with numerous modalities is an important direction of future work. Unfortunately, a lack of realistic datasets with numerous modalities is witnessed also in the multimodal VAE literature [2][6][9]. With our findings on the PolyMNIST dataset and our results on the proposed CUBICC dataset, as a realistic multimodal clustering benchmark, we hope to inspire future work to propose novel real-world benchmarks for clustering tasks with numerous modalities.
>
> We hope with this rebuttal we addressed the concerns from the Reviewer, possibly towards an adjustment of the score.
>
> **References:**
>
> [1] Hwang et al. Multi-view representation learning via total correlation objective. In NeurIPS, 2021. [2] Palumbo et al. MMVAE+: Enhancing the generative quality of multimodal VAEs without compromises. In ICLR, 2023. [3] Suzuki et al. Joint multimodal learning with deep generative models. arXiv preprint arXiv:1611.01891, 2017. [4] Vedantam et al. Generative models of visually grounded imagination. In ICLR, 2018. [5] Wu et al. Multimodal generative models for scalable weakly-supervised learning. In NeurIPS, 2018. [6]  Daunhawer et al. On the limitations of multimodal VAEs. In ICLR, 2022. [7] Alwassel et al. Self-supervised learning by cross-modal audio-video clustering. In NeurIPS, 2020. [8] Xu et al. Multi-vae: Learning disentangled view-common and view-peculiar visual representations for multi-view clustering. In ICCV, 2021. [9] Sutter et al. Generalized Multimodal ELBO. In ICLR, 2021.

---

> ### Author Response · Authors · 2023-11-22
> **Thank you for your feedback**
>
> We thank the Reviewer for the feedback and the suggestion, which we will take into account for the camera-ready version.

---

### Official Review · Reviewer_i4Eb · 2023-10-31

**Soundness:** 3 good
**Presentation:** 4 excellent
**Contribution:** 3 good
**Rating:** 6
**Confidence:** 4

**Summary:**

This work concerns multimodal VAE applied to clustering. In doing so the authors propose a three-fold approach: a VAE objective for latent clustering, an entropy based cluster number selection, and Diffuse VAE integrated to the multimodal setting. Furthermore the authors present the experiments using fairly common tasks for multimodal VAE plus for weakly supervised clustering.

**Strengths:**

Overall, this paper is well-written and easy to read. It’s advocately positioning itself on the problem of clustering with multimodal VAE, and, through a neat combination of ideas that are slightly improved over previous works, it present a holistic solution that outperform previous works both in multimodal VAE and weakly supervised clustering.

Specifically, the objectives are heavily inspired by the previous works. The choice of selecting $k$ based on entropy is both intuitive
incorporation of DiffuseVAE is direct yet impactful. Despite these individual components, what stands out is the overall pipeline being orchestrated coherently . The authors deserve commendation for doing so.

The experimental results are satisfactory and in line with common setups for multimodal. However this could be improved (see the weakness below).

**Weaknesses:**

As this paper is positioned as multimodal VAE for clustering, a broader empirical comparison with weakly supervised model would adds greatly to its value.

While the current version compares the proposed method with  [1,2,3], there is an wide range of research in weakly supervised clustering, including but not limited to [4, 5]. Although these may not be latent models, they (with other works) demonstrates the big picture of  weakly supervised clustering and deserve comparison


Additionally, the quantitative results relegated to the Appendix are pivotal for a comprehensive grasp of the proposed work's empirical performance. It would be beneficial to include these findings in the main body of the text.

Furthermore, a lot of quantitative results in Appendix is actually crucial in understanding the empirical performance of the proposed work. It would be beneficial to include these them in the main body of the text.


---------

[1] Jiang et al. Variational deep embedding: an unsupervised and generative approach to clustering IJCAI 2017

[2] Caron et al. Deep clustering for unsupervised learning of visual features.  ECCV 2018

[3] Tain et al. Contrastive multiview coding ECCV 2020

[4] Oner et al. Weakly Supervised Clustering by Exploiting Unique Class Count ICLR 2020

[5] Chang et al.  Deep adaptive image clustering ICCR 2017

[6] Yang et al. Joint unsupervised learning of deep representations and image clusters CVPR 2016

**Questions:**

Can the proposed multimodal VAE be applied to more complex data (say datasets with higher resolution, like AFHQv2, FFHQ) and if so, how would this impact the visual quality of the results?

---

> ### Author Response · Authors · 2023-11-18
> **Reply to Reviewer i4Eb (Part 1)**
>
> We thank the Reviewer for praising the positioning of our work, and highlighting its effectiveness as a holistic approach, bringing value to both multimodal VAEs and weakly-supervised clustering areas of research.
>
> We address below each question and concern raised by the reviewer.
>
> > As this paper is positioned as multimodal VAE for clustering, a broader empirical comparison with weakly supervised model would adds greatly to its value.
>
> > While the current version compares the proposed method with [1,2,3], there is an wide range of research in weakly supervised clustering, including but not limited to [4, 5]. Although these may not be latent models, they (with other works) demonstrates the big picture of weakly supervised clustering and deserve comparison.
>
> We thank the Reviewer for raising this point, and in the updated draft of the manuscript we clarify the positioning of CMVAE in the realm of weakly-supervised clustering. In this work, when referring to weakly-supervised approaches, we refer precisely to models that leverage weak supervision in the form of multiple modalities as in relevant related work [1][2][3][4]. We acknowledge that, outside of the multimodal VAE literature, weakly-supervised clustering might also refer to other forms of weak supervision (i.e., coarse labels, pairwise similarities, etc). However, these fall outside the scope of our work and do not directly fit into our experimental comparisons. For instance, the work from Oner et al. (2020) [5] mentioned by the Reviewer is not applicable to our experimental settings, as we do not have Unique Class Counts for our data. Nevertheless, we have extended the Related work section to mention other forms of weak supervision in clustering methods for completeness. Finally, while these comparisons are not the focus of our work, we test the seminal approach referenced by the reviewer from Chang et al. [5] in comparison with our method on the PolyMNIST dataset. In particular, in the Tables below, we report the results for DAC (Chang et al. [6]) trained on the five PolyMNIST modalities and compare the best performance across modalities with the performance of CMVAE. Note that, in line with other clustering approaches we test in our paper (see Appendix E.2.1), DAC achieves the best performance when trained on modality 3. This comparison once again validates the effectiveness of our approach compared to previously proposed clustering methods.
>
>
>  | DAC performance on PolyMNIST |               |               |               |
> |-------------------------------------------|---------------|---------------|---------------|
> |                                           | NMI           | ARI           | ACC           |
> | modality 0                                | 0.092 (0.018) | 0.040(0.009)  | 0.201 (0.007) |
> | modality 1                                | 0.212 (0.060) | 0.131 (0.052) | 0.300 (0.047) |
> | modality 2                                | 0.134 (0.045) | 0.070 (0.027) | 0.214 (0.029) |
> | modality 3                                | 0.724 (0.129) | 0.660 (0.154) | 0.781 (0.121) |
> | modality 4                                | 0.029 (0.013) | 0.011 (0.007) | 0.151 (0.016) |
> | Best (modality 3)                         | 0.724 (0.129) | 0.660 (0.154) | 0.781 (0.121) |
>
> |            | NMI           | ARI           | ACC           | K not needed for training |
> |------------|---------------|---------------|---------------|---------------------------|
> | DAC (best modality) | 0.724 (0.129) | 0.660 (0.154) | 0.781 (0.121) | no                        |
> | **CMVAE (ours)**     | 0.97 (0.02)   | 0.97 (0.02)   | 0.99 (0.01)   | yes                       |
>
>
>
> > Furthermore, a lot of quantitative results in Appendix is actually crucial in understanding the empirical performance of the proposed work. It would be beneficial to include these them in the main body of the text.
>
> We appreciate that the Reviewer recognizes the significance of our showcased quantitative results in the Appendix. We updated the manuscript to mention Table 3 in the main text, as these results show that CMVAE can effectively exploit a larger number of modalities in enhancing the clustering performance. While we made an effort to refer to significant Appendix results in the main text, we tried to avoid compromising the clarity of the manuscript, leaving thorough baseline results in the Appendix. However, we encourage the Reviewer to point out if there are specific quantitative results in the Appendix that the Reviewer feels should be part of the main text so that we can do out best to include them in the camera-ready version of the manuscript.
>
> **[Note: this reply continues in a separate comment.]**

---

> ### Author Response · Authors · 2023-11-18
> **Reply to Reviewer i4Eb (Part 2)**
>
> > Can the proposed multimodal VAE be applied to more complex data (say datasets with higher resolution, like AFHQv2, FFHQ) and if so, how would this impact the visual quality of the results?
>
> The proposed integration of DiffuseVAE [7] in the CMVAE model takes a significant step in the application of multimodal VAEs to realistic and complex settings (compare, for instance, our results on the CUBICC dataset to the highly related CUB Image-Captions dataset results in recent work [3][4]). DiffuseVAE has been successfully tested on images with a resolution of up to 256x256, and therefore, it would also work similarly in our scenario. With higher-resolution images, we argue that the VAE-based model would need to substantially increase the number of parameters to generate meaningful conditioning images to the DDPM. This comes at the cost of an increase in computational power, training, and inference time. On the other hand, the conditional DDPM would require fewer time steps than standard DDPM, thus optimizing the speed vs quality tradeoff.
>
> We hope our answers in this rebuttal addressed the questions and resolved the concerns from the Reviewer.
>
> **References:**
>
> [1] Locatello et al. Weakly-Supervised Disentanglement Without Compromises. In ICML, 2020. [2] Daunhawer et al. Identifiability Results for Multimodal Contrastive Learning. In ICLR, 2023. [3] Daunhawer et al.On the limitations of multimodal VAEs. In ICLR, 2022. [4] Palumbo et al. MMVAE+: Enhancing the generative quality of multimodal VAEs without compromises. In ICLR, 2023. [5] Oner et al. Weakly Supervised Clustering by Exploiting Unique Class Count, ICLR 2020. [6] Chang et al. Deep adaptive image clustering, ICCV 2017 [7] Pandey et al. DiffuseVAE: Efficient, controllable and high-fidelity generation from low-dimensional latents. Transactions on Machine Learning Research, 2022.

---

> ### Author Response · Authors · 2023-11-22
> **Follow-up on the rebuttal**
>
> Dear Reviewer i4Eb,
>
> We hope our rebuttal was effective in addressing your concerns/questions. Given the end of the discussion period is approaching, we would like to ask if you still have any further concerns or questions, particularly as follow-up to our rebuttal.
>
> Thank you in advance for your time and consideration!

---

### Author Response · Authors · 2023-11-18
**Reply to all Reviewers**

Dear Reviewers,

Thank you all for the constructive and helpful feedback.

We greatly appreciate your positive feedback regarding the soundness of our approach, its empirical performance, as well as the clarity and presentation of our work. Most reviewers seem to agree that our work brings relevant contributions, which are backed up by sound experimental evidence, and are impactful for performance. We appreciate Reviewer i4Eb praising the positioning and effectiveness of our approach, as a holistic solution outperforming previous works both in multimodal VAEs and weakly-supervised clustering. We also acknowledge the suggestion of a broader comparison with existing weakly-supervised clustering models. We believe this to be greatly useful feedback, and discuss this point in the rebuttal. We appreciate Reviewer XPst's comments on the soundness and clarity of our work, while concerns are expressed on novelty, as well as on the evidence and motivation behind claiming our proposed approach is scalable to numerous modalities. We consider and address these points in the rebuttal. Finally, we are glad that Reviewer GVdd highlights the relevance of our contributions and experiments, as well as the novelty and effectiveness of our post-hoc cluster selection procedure. We are thankful for the review pointing out parts of our work that might benefit from further clarifications, and address the Reviewer's questions in the rebuttal.


In summary, in the rebuttal we address the following main concerns

**Reviewer i4Eb**: We further elaborate on the comparison between our approach and existing weakly-supervised clustering approaches. We also include a comparison with a work referenced by the Reviewer, that validates our proposed approach in comparison with existing methods in the realm of weak supervision for clustering tasks. Inspired by the comment from the Reviewer, we clarify that when referring to weak supervision in our paper, as in relevant related work [1][2][3][4],  we precisely refer to weak supervision in the form of co-occurring multiple data modalities. In the updated version of the draft, by clarifying this point, we position our work more clearly in the realm of weakly-supervised clustering methods.



**Reviewer XPst**: We address the concerns on novelty, further clarifying the novel aspects of our work and the relevance of our contributions. We clarify our claim of scalability of our method in the number of modalities. In doing so, we further elaborate on the significance of our showcased performance for multimodal clustering, including additional comparisons with previous multimodal clustering methods.

**Reviewer GVdd**: We address your questions, providing clarifications to certain parts of our paper.

[1] Locatello et al. Weakly-Supervised Disentanglement Without Compromises. In ICML, 2020. [2] Daunhawer et al. Identifiability Results for Multimodal Contrastive Learning. In ICLR, 2023. [3] Daunhawer et al. On the limitations of multimodal VAEs. In ICLR, 2022. [4] Palumbo et al. MMVAE+: Enhancing the generative quality of multimodal VAEs without compromises. In ICLR, 2023.

---

### Meta-Review · Area_Chair_zHTw · 2023-12-09

**Metareview:**

The paper proposes a multi-step algorithm for representation learning and coherent generation of multi-modal data:
1. Multi-view VAE, for which the author is mostly using MMVAE+ by Palumbo et al, 2023, with a discrete variable for modeling clusters/label structure, using the parameterization by Jiang et al, 2017,
2. post-processing of clusters to estimate number of classes,
3. improved generation with diffusion models, similar to Pandey et al 2022.
While in general the technical contribution to each step is not too high, the reviewers are satisfied with the orchestration/execution and the results.

**Justification For Why Not Higher Score:**

The overall technical novelty is not extremely high.

**Justification For Why Not Lower Score:**

Reviewers are generally happy with the experimental evaluation.

---

### Decision · Program_Chairs · 2024-01-16

Accept (poster)